# Junction-based lamellipodia drive endothelial cell rearrangements in vivo via a VE-cadherin-F-actin based oscillatory cell-cell interaction

Ilkka Paatero [1,2], Loïc Sauteur[1], Minkyoung Lee[1], Anne K. Lagendijk[3], Daniel Heutschi[1], Cora Wiesner[1], Camilo Guzmán[3], Dimitri Bieli[1], Benjamin M. Hogan[3], Markus Affolter[1] & Heinz-Georg Belting [1]

Angiogenesis and vascular remodeling are driven by extensive endothelial cell movements. Here, we present in vivo evidence that endothelial cell movements are associated with oscillating lamellipodia-like structures, which emerge from cell junctions in the direction of cell movements. High-resolution time-lapse imaging of these junction-based lamellipodia (JBL) shows dynamic and distinct deployment of junctional proteins, such as F-actin, VE-cadherin and ZO1, during JBL oscillations. Upon initiation, F-actin and VE-cadherin are broadly distributed within JBL, whereas ZO1 remains at cell junctions. Subsequently, a new junction is formed at the front of the JBL, which then merges with the proximal junction. Rac1 inhibition interferes with JBL oscillations and disrupts cell elongation—similar to a truncation in *ve-cadherin* preventing VE-cad/F-actin interaction. Taken together, our observations suggest an oscillating ratchet-like mechanism, which is used by endothelial cells to move over each other and thus provides the physical means for cell rearrangements.

[1] Department of Cell Biology, Biozentrum, University of Basel, Basel 4056, Switzerland. [2] Turku Centre for Biotechnology, University of Turku and Åbo Akademi University, Turku 20520, Finland. [3] Division of Genomics of Development and Disease, Institute for Molecular Bioscience, The University of Queensland, St Lucia, QLD 4072, Australia. These authors contributed equally: Loïc Sauteur, Minkyoung Lee. Correspondence and requests for materials should be addressed to M.A. (email: markus.affolter@unibas.ch) or to H.-G.B. (email: heinz-georg.belting@unibas.ch)

Organ morphogenesis is driven by a wealth of tightly orchestrated cellular behaviors, which ensure proper organ assembly and function. The cardiovascular system is one of the most ramified vertebrate organs and is characterized by an extraordinary plasticity. It forms during early embryonic development, and it expands and remodels to adapt to the needs of the growing embryo. In adult life, this plasticity allows flexible responses, for example, during inflammation and wound healing[1,2].

At the cellular level, blood vessel morphogenesis and remodeling are accomplished by endothelial cell behaviors including cell migration, cell rearrangement and cell shape changes[3–5]. This repertoire of dynamic behaviors allows endothelial cells to rapidly respond to different contextual cues, for example during angiogenic sprouting, anastomosis, diapedesis or regeneration. In particular, it has been shown that endothelial cells are very motile, not only during sprouting, but also within established vessels, where they migrate against the blood flow[6,7].

Endothelial cell migration has been extensively studied in different in vivo and in vitro systems mainly focusing on angiogenic tip cell behavior and the interaction of endothelial cells with the extracellular matrix (ECM)[8,9]. However, endothelial cells can also shuffle positions within an angiogenic sprout[10], and these cellular rearrangements require the junctional adhesion protein VE-cadherin/CDH5[11–13]. Moreover, in vivo analyses in avian and fish embryos have shown that endothelial cells can migrate within patent blood vessels emphasizing that regulation of endothelial cell adhesion and motility is critical during vascular remodeling processes[6,7,14,15].

Although many aspects of sprouting angiogenesis and vascular remodeling rely on endothelial cell interactions[3], the exact role of endothelial cell junctions (and in particular that of VE-cad) in these processes is not well understood. Indeed, rather than supporting an active function for VE-cad in dynamic cell behaviors, most studies point to a restrictive or permissive role, consistent with the maintenance of endothelial integrity[16–18]. On the other hand, the observation that loss of VE-cad function can inhibit cell rearrangements suggests an active contribution to this process[12,13].

To decipher the cellular and molecular mechanisms, which enable cells to move within the endothelium, we have focused on the process of anastomosis during the formation of the dorsal longitudinal anastomotic vessel (DLAV) in the zebrafish embryo by high-resolution time-lapse microscopy. This process occurs in a relatively stereotypical manner and involves a convergence movement of endothelial cells, which is illustrated by extensive cell junction elongation[19]. Ultimately, this process alters tube architecture and converts unicellular vessels to multicellular vessels[20]. By in vivo time-lapse imaging of several junctional components and pharmacological interference with F-actin dynamics, we are able to describe a actin-based mechanism, which allows endothelial cells to move along each other while maintaining junctional integrity. In particular, we describe a rearrangement mechanism, which is initiated by junction-based lamellipodia (JBL) leading to the formation of distal, VE-cad based attachment sites, which in turn serve as an anchor point for junction elongation. We propose that the oscillating behavior of JBL, which depends on F-actin polymerization as well as contractility, provides a general mechanism of endothelial cell movement during blood vessel formation and vascular remodeling.

## Results

### Changes of vessel architecture during blood vessel formation.
Blood vessel formation is associated with prominent cell shape changes and cell rearrangements. The DLAV presents a well-defined in vivo model to analyze how a wide repertoire of endothelial cell activities leads to the formation of a new blood vessel, starting with establishment of an interendothelial contact point, followed by the formation of a continuous luminal surface and the transformation from a unicellular to a multicellular tubular architecture. Unicellular and multicellular tubes are easily discerned by junctional patterns, whereas unicellular tubes display isolated rings separated by segments without any junction, multicellular tubes have a continuous network of multiple junctions along their longitudinal axis (Fig. 1a). To gain more insight into this transformation process, we used a reporter line expressing a full-length VE-cadherin fluorescent protein fusion (VE-cad-Venus)[21] (Fig. 1b–e, Supplementary Movie 1) and performed in vivo time-lapse experiments between 27 and 40 h post fertilization (hpf). These experiments showed that most DLAVs were initially unicellular tubes, and that the majority (69%, $n = 26$ (8 embryos)) of DLAV segments were transformed to a multicellular configuration before 40 hpf (Fig. 1b–e, Supplementary Fig. 1a). The transformation from the initial tip cell contact to a multicellular vessel, with a continuous cell–cell junction network, took several hours (median 190 min, segments $n = 14$ (8 embryos), Supplementary Fig.1b), with high variability between individual segments. During this time window, the endothelial cell–cell junctions expanded extensively from initial spot-like structures to elongated junctions covering the entire DLAV segment. However, movement of the junctions was also seen in perfused vessels (Supplementary Fig. 1c). The cellular rearrangements are thus occurring both in nascent non-lumenized vessels and also in inflated, perfused vessels.

**The thickness of remodeling junctions is polarized**. When we analyzed ring-shaped junctions in the DLAV of VE-cad-Venus embryos in more detail, we observed that the junctions were not uniform in thickness along their circumference in unicellular vessels. In medial regions, the junctions formed significantly thicker, and more diffuse, pattern than on the lateral sides (Fig. 1f, j), coinciding with the general direction of endothelial cell movements during anastomosis. In contrast, we did not observe such junctional polarity in multicellular vessels (Fig. 1j). We confirmed these observations by immunostainings for the junctional proteins VE-cadherin and ZO1, which showed that in the newly formed junctions in unicellular configuration, the medial junctional domains were consistently thicker than the lateral domains (Fig. 1g–i, k). Again, this polarity of junctional thickness was not seen in vessel areas of more mature multicellular architecture (Fig. 1k).

**Remodeling junctions form junction-based lamellipodia**. To gain insight into the nature of this junctional polarity, we performed live-imaging experiments on Cdh5-Venus expressing transgenic embryos at high temporal resolution. Here, we observed that the polarized junctional thickenings are formed by dynamic lamellipodia-like protrusions (Fig. 2a, b, Supplementary Movie 2). In addition, we used a F-actin visualizing EGFP-UCHD transgenic fish line in a similar setup (Fig. 2c, d, Supplementary Movie 3). Remarkably, F-actin and VE-cadherin, both showed similar oscillatory dynamics with a median duration of 6 min (Fig. 2e). Moreover, the protrusions were oriented along the vessel axis (Fig. 2f), which is consistent with the increased junctional thickness of medial junctional domains.

To test whether this dynamic junctional behavior is restricted to the process of anastomosis or is a more general behavior, we further analyzed the F-actin fluctuations during junctional remodeling in the dorsal aorta. Here, we found that the relative intensity of EGFP-UCHD was increased at the site of forming protrusions (Fig. 3a, b, Supplementary Movie 4), indicating

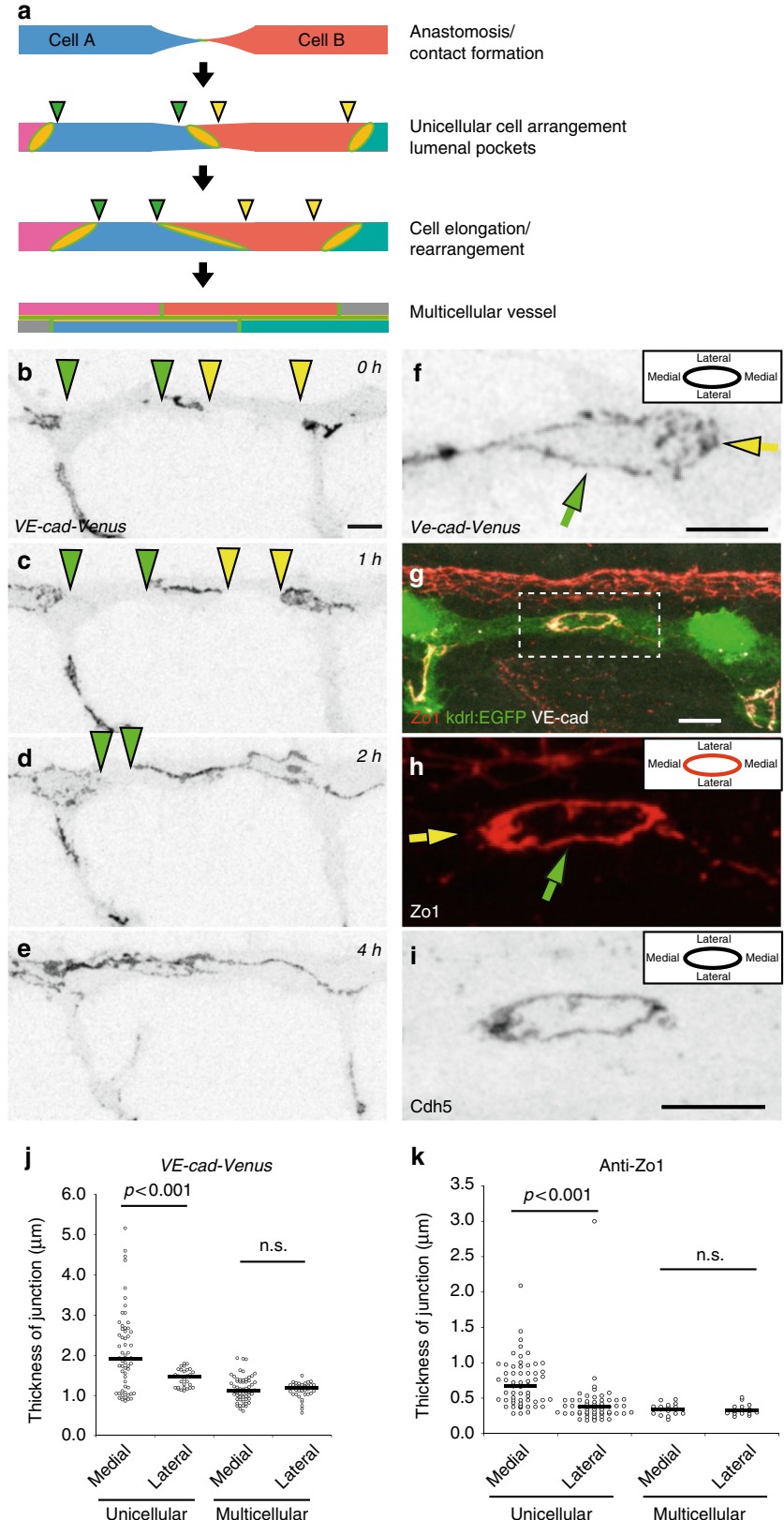

recruitment of additional F-actin. We further analyzed the dynamic behavior of junctional F-actin using kymographs (Fig. 3c, d) and found that the fluctuations in intensity occurred in rather rhythmic patterns, and at the same junctional site, protrusions were generated repeatedly within regular intervals,

indicative for oscillations in F-actin intensity in the remodeling junction.

The polarized occurrence and directionality of protrusions along the direction of vessel growth suggests that they are involved in endothelial cell movements. To address this, we

**Fig. 1** Polarized thickness in remodeling junctions. **a** Schematic model of cellular rearrangements during formation and maturation of dorsal longitudinal anastomosing vessel (DLAV) in zebrafish. At first, the tip cells (Cell A and B) of two anastomosing endothelial sprouts contact and form a de novo junction. Next, a vessel with unicellular architecture is formed, where junctions are not interconnected but are visible as separate rings. Then, through cellular rearrangement and elongation, the junctions elongate along vessel axis until junctions are interconnected and vessel reaches the final multicellular architecture. The edges of the junctional gaps in the unicellular vessels are marked with green and yellow triangles. **b–e** Still pictures of a time-lapse movie (Supplementary Movie 1) showing EC junctions of *Tg*(BAC(*cdh5:cdh5-ts*)) embryo, which expresses VE-cad-Venus fusion protein, during transition from unicellular to multicellular vessel during DLAV formation, in inversed contrast starting around 28 hpf. The edges of the junctional gaps in the unicellular vessels are marked with green and yellow triangles. **f** Close-up image of VE-cad-Venus embryos. The diffuse thickening of medial domain of the junction is marked with a yellow arrow. **g–i** Whole-mount immunofluorescence staining of the DLAV using anti-ZO1, anti-VE-cad (rat) of 28–30 hpf (*Tg*(*kdrl: EGFP^{s843}*)) embryos. **h** The yellow arrow points to the medial junctional domain and the green arrow to the lateral junctional domain. **j** Quantification of junctional thickness measurements. $n = 44$ unicellular junctions and 48 multicellular junctions from 5 VE-cad-Venus *Tg*(BAC(*cdh5:cdh5-ts*)) embryos. Non-parametric Kruskal–Wallis test was used. **k** Quantification of the junctional thickness measurements from immunostainings of *Tg*(*kdrl:EGFP^{s843}*) embryos. $n = 58$ unicellular junctions and 17 multicellular junctions from 8 embryos. Non-parametric Kruskal–Wallis test was used. Scale bars 10 μm

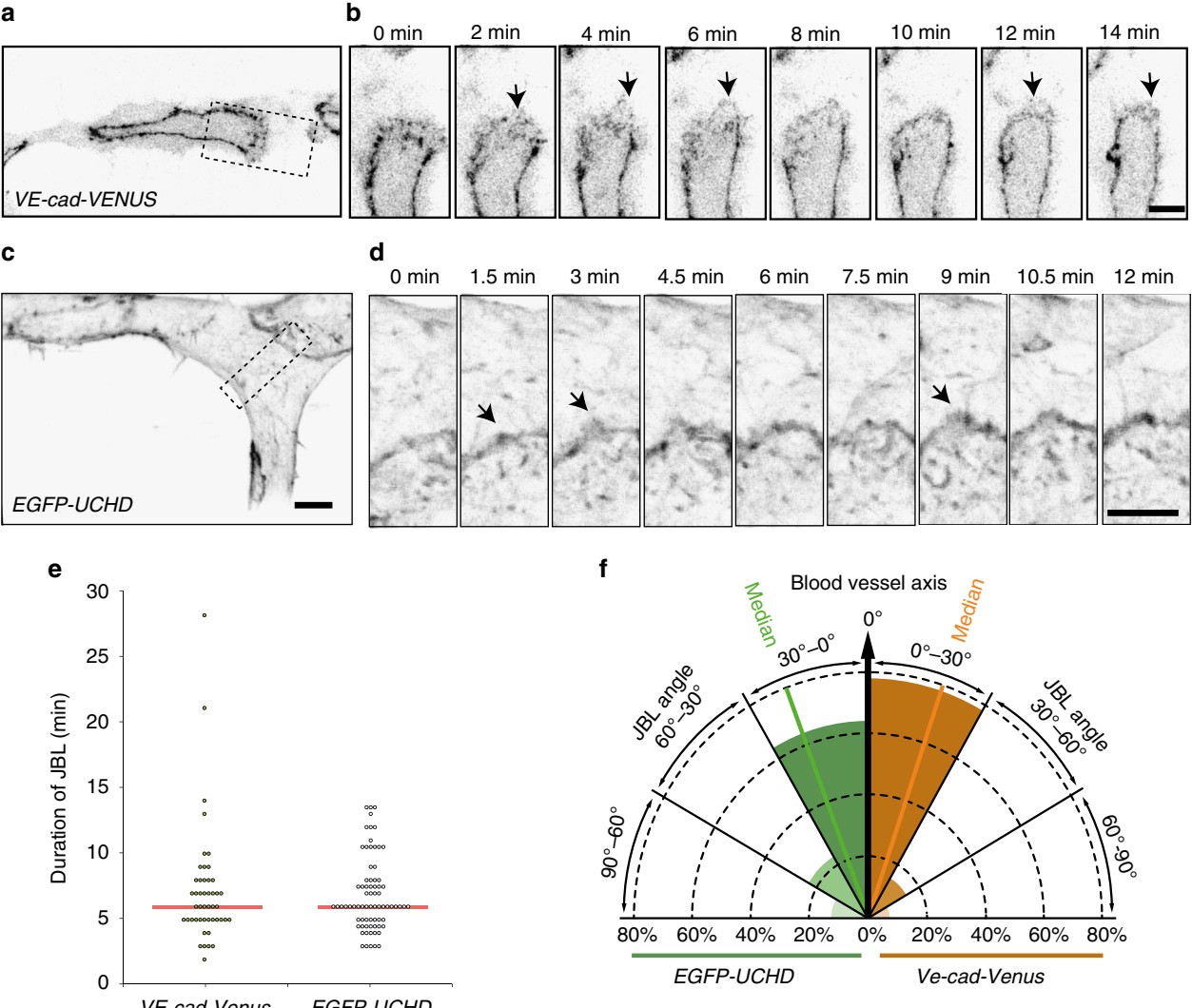

**Fig. 2** Active VE-cad and F-actin behavior of junction-based lamellipodia. **a, b** Still images from a movie (Supplementary Movie 2) of a VE-cad-Venus expressing embryo *Tg*(BAC(*cdh5:cdh5-ts*)), showing the DLAV at 30 hpf in inversed contrast. **b** A magnification of the inset in **a**. Arrows point to JBL. **c, d** Still images from a movie (Supplementary Movie 3) of a EGFP-UCHD expressing embryo (*Tg*(*fli:Gal4ff^{ubs3}*, *UAS:EGFP-UCHD^{ubs18}*)) showing the DLAV at 30 hpf in inversed contrast. **d** A magnification of the inset in **c**. Arrows point to JBL. **e** Scatter plot of quantitation of the duration of the JBL with the VE-cad-Venus transgene ($n = 48$ in 6 embryos) and EGFP-UCHD movies ($n = 74$ in 6 embryos), red line represents the median. **f** Quantitation of JBL angle in the DLAV in respect to the blood vessel axis (0°) using the EGFP-UCHD transgene ($n = 103$ from 6 embryos) or Cdh5-Venus transgene ($n = 41$ from 5 embryos). Scale bars 5 μm

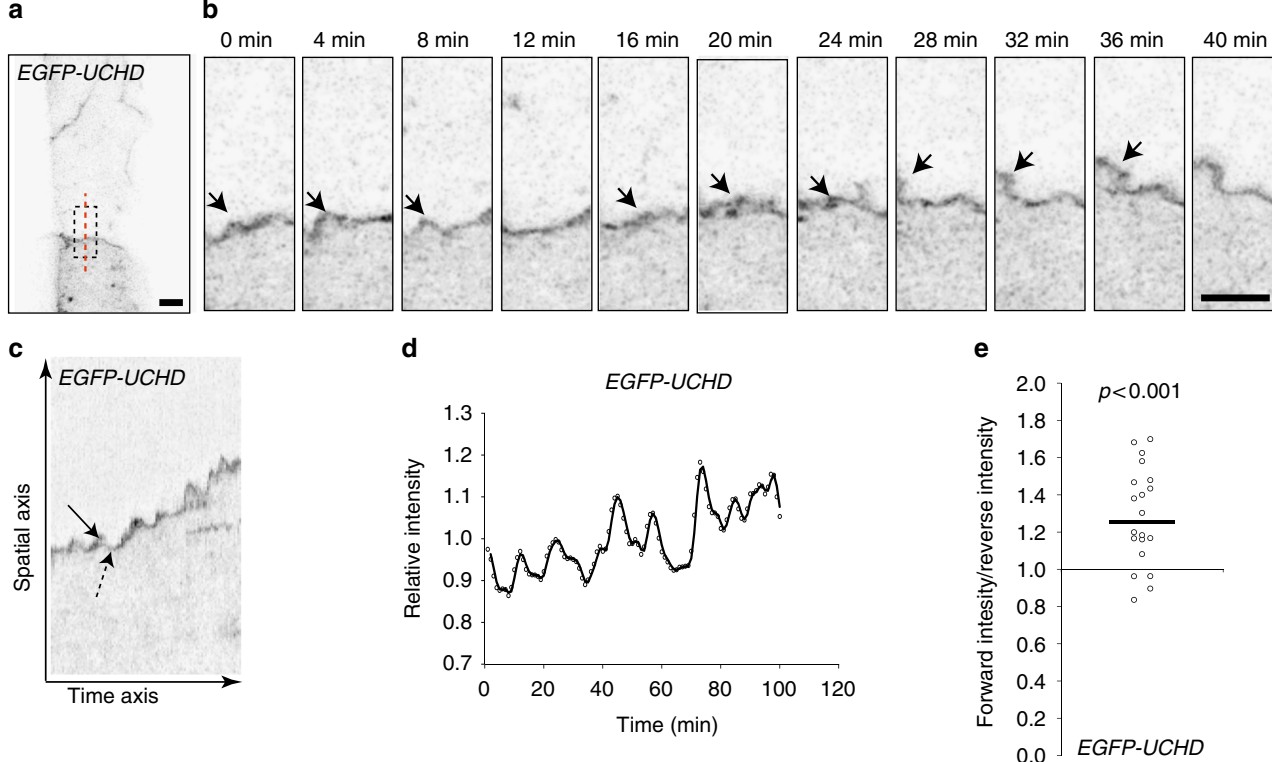

**Fig. 3** Oscillatory F-actin dynamics during remodeling of cell–cell junctions. **a**, **b** Still images from a movie (Supplementary Movie 4) showing JBL formation in the dorsal aorta of an EGFP-UCHD expressing 2dpf embryo (*Tg(fli:Gal4ff^{ubs3}; UAS:EGFP-UCHD^{ubs18})*), shown in inversed contrast. **b** A magnification of the inset in **a** and the red dashed line indicates the site for kymograph in **c**. Arrows point to a JBL, seen as a local thickening of the junction. **c** Kymograph across the junction. Solid arrow denotes forward movement and dashed arrow backward movement of the junction. **d** Intensity plotting of a EGFP-UCHD JBL kymograph. **e** Scatter plot of the relative EGFP-UCHD intensity during forward and backward movements (*n* = 20 events, 4 movies). EGFP-UCHD intensity value in a forward movement was divided with intensity value during subsequent reverse movement. Non-parametric one sample Wilcoxon signed rank test was used as statistical test. Scale bars 5 μm

analyzed the potential association between local junctional movements and the occurrence of F-actin protrusions in the dorsal aorta (Fig. 3c, d). Analysis of F-actin intensities showed that higher intensities were associated with local forward movement of junctions than with reverse movement (Fig. 3e).

Taken together, we observe an F-actin-based protrusive endothelial behavior, which occurs during junctional remodeling in vivo. Because of their similarity to 'classical' lamellipodia, their oscillating behavior and structural connection with endothelial cell junctions, we call these protrusions junction-based lamellipodia or JBL.

**JBL form at the front end of elongating junctions**. Blood vessel anastomosis is driven by the convergence movement of two tip cells and is associated with an elongation of their mutual cell junction. The formation of JBL at the junctional poles suggested that these dynamic structures may generate tractive forces, which contribute to junction elongation. However, cell junctions demarcate the interface between two cells and our above analyses did not differentiate, whether cells form JBL at their respective junctional front or rear ends or both. To analyze the contributions of individual cells to F-actin protrusions, we generated a transgenic zebrafish line expressing a photoconvertible mCLAV-UCHD fusion protein in endothelial cells. For mosaic analysis, we photoconverted mCLAV-UCHD in single SeA sprouts 2–3 h prior to contact initiation between tip cells from neighboring segments and recorded the behavior of the junctional F-actin later during anastomosis and formation of the DLAV (Fig. 4a). In vivo photoconversion resulted in efficient green-to-red conversion

(Fig. 4b), allowing the analysis of differentially labeled F-actin pools during DLAV formation (Fig. 4c). Photoconverted and non-converted mCLAV-UCHD-labeled endothelial cell junctions and showed matching patterns within the anastomotic junctional ring—except for the poles of the anastomotic ring where F-actin based protrusions were forming. Here, the colocalization of the two markers was suspended and we found elevated levels of either of the two F-actin pools at the front end of the junction with respect to cell movement (green JBL over red cell, or vice versa, 24 out of 28 events, *p* < 0.001). This oriented localization is consistent with an involvement of JBL in the forward movement of tip cells during anastomotic convergence movements.

**F-actin protrusions precede junctional movements**. Since the above observations suggest a step-wise mechanism of cell–cell interaction during JBL function, we set out to explore the spatio-temporal relationship between F-actin dynamics and the dynamics of other junctional components. To this end we generated transgenic fish lines expressing red F-actin (mRuby2-UCHD), which allows a direct comparison with other fluorescently labeled junctional components (e.g., EGFP-ZO1 and VE-cad-Venus) (Fig. 5a, b and c, d, respectively; Supplementary movies 5 and 6). Both VE-cad-Venus and EGFP-ZO1 followed the junctional F-actin front (11 and 9 movies analyzed, respectively); however, a different distribution pattern was observed during JBL formation. VE-cadherin localized diffusely at the front, largely overlapping with the F-actin protrusions (Fig. 5b, 60–120 s.). In contrast, EGFP-ZO1 showed a more defined localization and initially remained associated with the junction at the

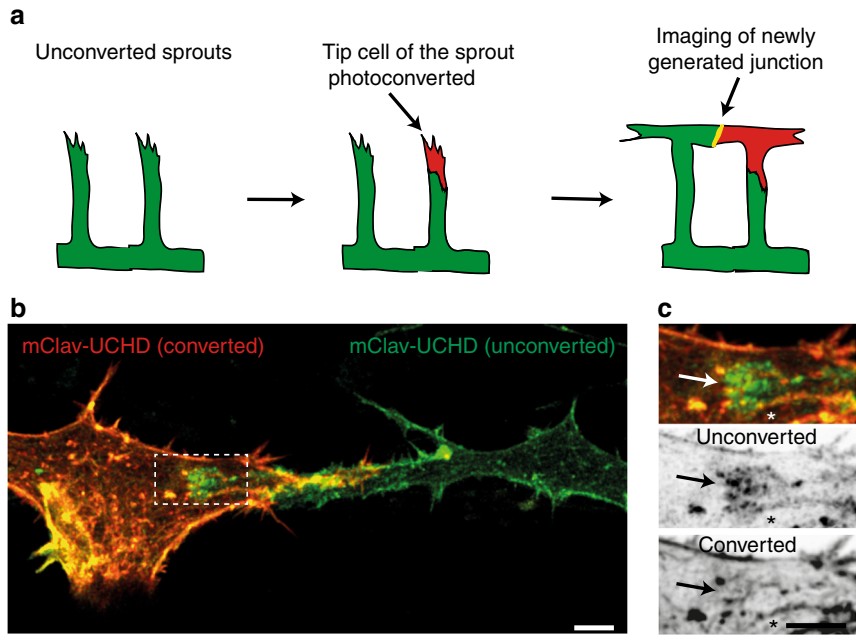

**Fig. 4** JBL formation at the distal tip of the junction during DLAV anastomosis. **a** Schematic representation of the mClav2-UCHD photoconversion experiment. **b** Image of photoconverted and unconverted mClav2-UCHD cells in the DLAV of an *Tg(fli:Gal4ff^{ubs3};UAS:mClav2-UCHD^{ubs27})* embryo, at 32 hpf. **c** A close up of the inset in **b**. Arrows point to differentially labeled JBL and asterisk (*) marks the junction outside JBL. Scale bar 5 μm

proximal end of the protrusion (Fig. 5d, 0–36 s.). However, at later time points (Fig. 5d, 72 s.), we observed EGFP-ZO1 accumulation also at the front edge of the JBL, indicating the formation of a new junction at this site (Fig. 5d). To directly differentiate the distribution of VE-cadherin and ZO1, we injected a mCherry-ZO1 encoding plasmid into VE-cad-Venus recipients. The differential localization of both proteins confirmed our previous observations and showed that ZO1 distribution is largely restricted to cell junctions. In contrast, VE-cad was also found within areas outside of these junctions (Fig. 5e, f, Supplementary Movie 7).

Therefore, the respective distribution of ZO1 and VE-cad represent different aspects of JBL formation and illustrates a stepwise mechanism of JBL function. First, F-actin based JBL emanate from EC junctions, which are maintained. The JBL contains diffusely distributed VE-cad. This population of VE-cad precedes formation of the new junction in front of the JBL and may therefore provide adhesive properties for the JBL prior to formation of the new junction. Interestingly, a gradual movement of the old junction towards the new junction was observed in the EGFP-ZO1 movies (Fig. 5d). This indicates that the proximal junction is not resolved in situ, but is actually pulled forward and eventually merges with the distal junction.

**F-actin is required for JBL formation and junction elongation.** To elucidate the molecular mechanism underlying JBL function during endothelial cell movements, we examined the requirement of F-actin dynamics by pharmacological interference. Latrunculin B and NSC23766 (a Rac1 inhibitor) are potent inhibitors of F-actin polymerization and lamellipodial F-actin remodeling, respectively[22,23]. We used acute treatments to avoid secondary effects and performed live-imaging on rearranging endothelial cell junctions. Inhibition of F-actin polymerization led to pronounced defects in JBL formation. In 5 movies, we observed only 13 JBL, compared to 50 JBL in control embryos (Fig. 6a, b, Supplementary Movies 8 and 9). Moreover, those JBL, which did

form in the presence of Latrunculin B, lasted longer, indicating additional defects in lamellipodial dynamics. Inhibition of Rac1 did not interfere as strongly with JBL formation as inhibition of F-actin polymerization. Here, we observed 49 JBL in 10 movies. However, these JBL displayed prolonged duration indicating defects in JBL dynamics (Fig. 6a, b, Supplementary Movie 10).

To test whether interfering with F-actin and JBL dynamics has consequences for endothelial cell rearrangements, we analyzed the effect of Latrunculin B and NSC23766 on junction elongation. Both compounds inhibited the elongation of the junction during DLAV formation (Fig. 6c, d) indicating that proper function of JBL is necessary for junctional elongation during anastomosis.

Although the gross morphology of JBL—based on VE-cad-Venus—looked relatively normal upon Rac1 inhibition, we wondered whether localization of other junctional proteins was affected. During the late phase of JBL oscillation, ZO1 is often localized in two lines outlining the distal (new) and proximal (old) junctions (Fig. 5d). When we compared EGFP-ZO1 distribution in control and NSC23766 treated embryos, we found that distal junctions were forming, showing that Rac1 inhibition did not abrogate de novo junction formation (Fig. 6e). Furthermore, despite the appearance of double junctions, we still did not observe any junction elongation in these instances.

As the above results suggested that Rac1 is primarily involved in the regulation of JBL dynamics rather than their structural properties, we analyzed the dynamics of junctional F-actin intensity oscillations (Supplementary Fig. 3b-d). In control embryos we observed a periodic pattern of JBL dynamics, but in NSC23766-treated embryos the periodicity was reduced (Supplementary Fig. 3e), indicating that junctional F-actin oscillations become less coordinated upon Rac1 inhibition.

**VE-cadherin/F-actin interaction is required for JBL function.** Next, we wanted to explore the role of VE-cad in JBL function, because several lines of evidence suggested that VE-cad may play

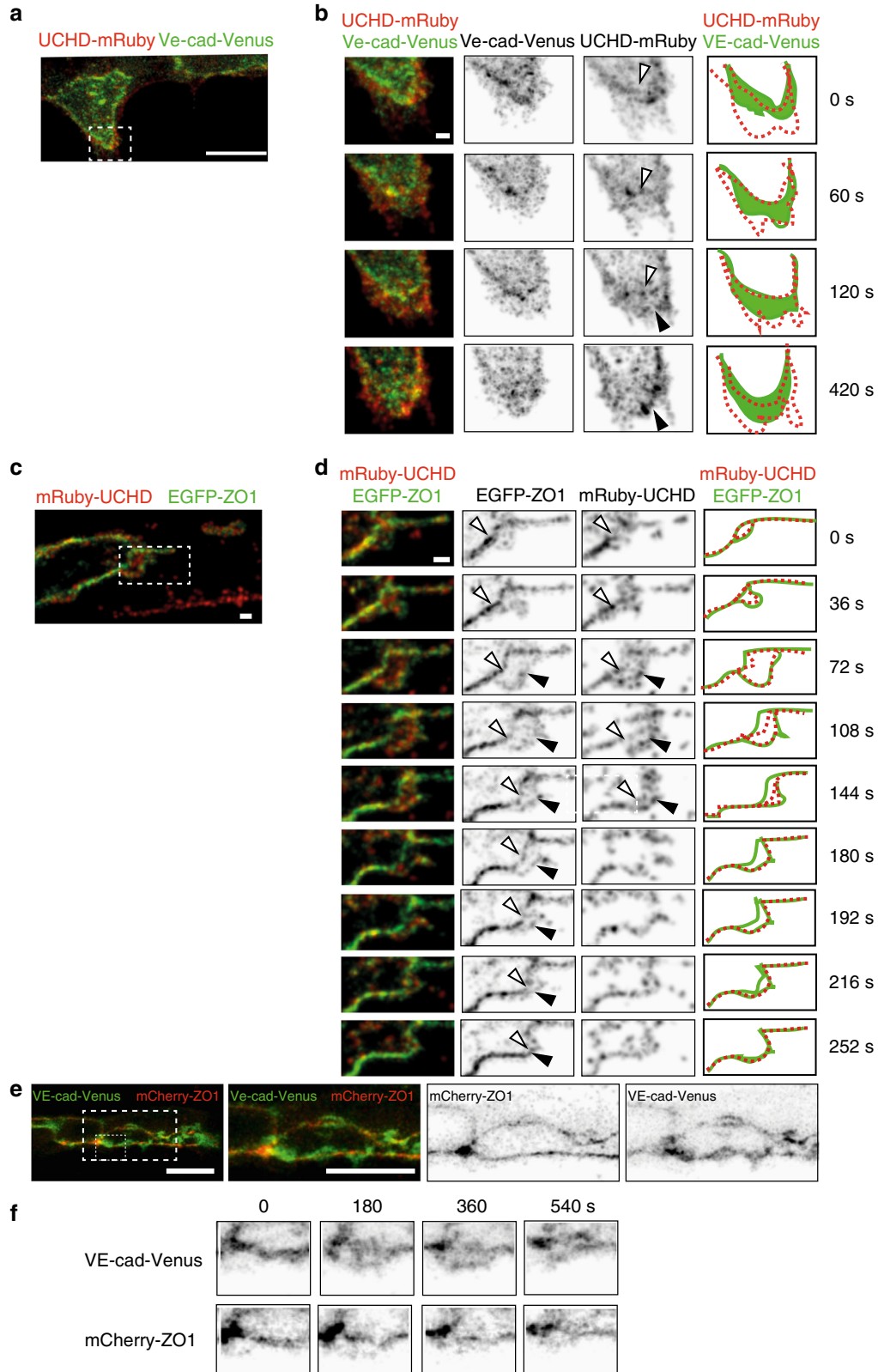

an important role in this process. In ve-cadherin zebrafish null mutants (*ve-cad/cdh5*[ubs8]), blood vessel architecture is generally disrupted and discontinuous lumens prohibit blood flow[13,24]. These defects are caused by an inability of mutant ECs to perform coordinated cell junction elongation, which is required for multicellular tube formation during angiogenesis. Furthermore, the

*ve-cad* mutant defects in junction elongation can be copied by the inhibition of F-actin polymerization[13].

Our finding that VE-cad accumulates in JBL prior to junction formation is consistent with the above observations, and collectively, they indicate a functional interaction between VE-cad and F-acting during junction elongation. To address this

**Fig. 5** Distinct dynamics of VE-cadherin, F-actin and ZO1 during JBL formation. **a, b** Still images (Supplementary Movie 5) of an embryo showing the DLAV around 32 hpf in an embryo expressing both mRuby2-UCHD and VE-cad-Venus *Tg(fli:Gal4ff^ubs3;UAS:mRuby2-UCHD^ubs20;BAC(cdh5:cdh5-Venus))*. **b** A time series magnification of the inset in **a**. Individual channels are shown in inversed contrast. Similar observations were made in 11 movies. Open arrow head points to established junctions and black arrowhead to pioneering junction. **c** and **d**) Still images of an embryo showing DLAV around 32 hpf (Supplementary Movie 5) in an embryo expressing EGFP-ZO1 and mRuby2-UCHD (*Tg(fli:Gal4ff^ubs3;UAS:mRuby2-UCHD^ubs20;UAS:EGFP-hZO1^ubs5)*). Imaged at rate of 12 s/stack. Similar observations were made in 9 movies. Open arrow head points to established junctions and black arrowhead to pioneering junction. **e** Images of endothelial cells in a VE-cad-Venus expressing embryo injected with mCherry-ZO1 encoding plasmid *Tg(BAC(cdh5:cdh5-ts)); fli1ep: mCherry-ZO1))* (*n* = 7 embryos). **f** Close-up from panel **e**. Both channels are shown in inverted contrast. Scale bars 1 μm (**b–d**) and 10 μm (**a, e**)

possibility, we generated a targeted mutation in *ve-cad* (*cdh5^ubs25*), which results in a "tailless" protein, lacking a portion of the cytoplasmic domain of VE-cad including the ß-catenin binding site, essential for VE-cad/F-actin interaction (Supplementary Fig. 4).

Despite the deletion, the VE-cadherin protein correctly localizes in the endothelial cell–cell junctions (Fig. 7h, k and Supplementary Fig. 4e). Homozygous *ve-cad^ubs25* mutants are embryonic lethal and display phenotypes similar to those of null mutants, including tip cell/stalk cell dissociation and defective blood circulation (Supplementary Fig. 5a-e)[13]. However, some of the defects are less pronounced and we observed blood flow in the DA and—in rare cases—in the DLAV (Supplementary Fig. 5a, Supplementary Movie 11). This shows that the extracellular domain of VE-cad mediates some inter-endothelial adhesion, which leads to a hypomorphic phenotype.

To assess the requirement for VE-cadherin/F-actin interaction for JBL formation, we first examined whether the junctional rings of VE-cad truncation (*cdh5^ubs25*) mutants displayed polarized thickness during anastomosis. Immunofluorescent staining for ZO1 revealed that medial junctions of mutants were narrower compared to heterozygotes, while the thickness of the lateral sides was not affected (Fig. 7a, b). We then tested whether the dynamics of F-actin protrusions were affected in VE-cad truncation (*cdh5^ubs25*) mutants. Time-lapse analyses in embryos expressing EGFP-UCHD indicated that F-actin protrusions oscillated more slowly in *cdh5^ubs25* mutants compared to wild-types in a manner similar to how protrusions behaved in the presence of the Rac1 inhibitor (Fig. 7c). Furthermore, by measuring the time-intervals between the end and the beginning of a JBL cycle, we found that this "lag phase" is more than doubled (Supplementary Fig. 4f) suggesting that full-length VE-cad is required for the initiation of JBL formation.

To test whether these defects in JBL have consequences for vascular morphogenesis, we examined the junctional architecture of forming SeA. In *cdh5^ubs25* mutants, we observed increased interjunctional gaps, which indicate a defect in multicellular tube formation due to a failure in junction elongation (Fig. 7d–l). We also compared relative nuclear movements, which are associated with stalk cell elongation. Consistent with the observed defects in junctional rearrangement, *cdh5^ubs25* mutants displayed significantly slower rearrangement of EC cell nuclei compared to wild-types (Fig. 7m, n). Taken together, these findings show that VE-cadherin plays an important role in F-actin dynamics and that the VE-cadherin/F-actin interaction is essential for JBL function, junction elongation and endothelial cell rearrangements in vivo.

## Discussion

In this study, we have investigated the mechanisms by which junctional dynamics contribute to endothelial cell movements during blood vessel formation in vivo.

By time-lapse imaging of different structural components of endothelial cell junctions, we observe a dynamic and differential deployment of these proteins during junctional remodeling, which leads to the formation of transient lamellipodia-like

protrusions, that we call JBL. Together with our analyses of F-actin dynamics and VE-cad function, our findings suggest a mechanism of cellular and molecular interactions, which allows endothelial cells to use each other as adhesive substrates and for force transmission during cell migration and elongation (Fig. 8). In essence, JBL act by a ratchet-like mechanism, which consists of F-actin-based protrusions and VE-cad based interendothelial cell adhesion. While F-actin protrusions provide the motive force, VE-cad based adhesion serves as an intercellular clutch.

Our time-lapse experiments show that junction elongation is associated with oscillating JBL, which occur at a frequency of about one every 6 min. This oscillatory behavior, their polarized localization at the leading edge of the junction, their role in cell movements together with their dependency on F-actin polymerization as well as Rac1 GTPase activity indicates that these protrusions share a functional basis with "classical" lamellipodia[25]. However, and in contrast with "classical" lamellipodia, the protrusions, which we are describing, emanate from interendothelial cell junctions and require VE-cad for adhesion and force transmission.

Here, we describe the role of JBL during angiogenesis in the zebrafish, where they promote endothelial cell elongation and rearrangements. It remains to be seen, whether JBL are unique to endothelial cells or employed more widely during morphogenetic processes of different cell types. Cadherin-based cell interactions have been shown to be essential for dynamic cell movements in several morphogenetic processes (reviewed by refs. [26,27]). In some aspects, the mechanism of JBL function appears to be similar to the one described for border cell migration during *Drosophila* oogenesis[28]. Here, actin-based cell protrusions and E-cadherin interactions between border and nurse cells are thought to be important to form an anchor point at the leading edge of border cells.

Oscillatory junctional protrusions of endothelial cells have also been described in cultured HUVECs[29,30]. Here, so-called junction-associated intermittent lamellipodia (JAIL) form in similar intervals as JBL. However, several characteristic differences suggest that JAIL and JBL represent different cellular activities. As the name indicates, JAIL formation is preceded by the local dissolution of the existing junction, which is thought to trigger the formation of an actin-based protrusion followed by the formation and stabilization of a new junction. In contrast, JBL formation is not associated with the dissolution of an existing junction. This leads to the formation of a characteristic double junction (proximal and distal) at the pole of the junctional ring (Fig. 8). Conceptually, maintenance of cell junctions should be a prerequisite for endothelial cell rearrangements in a perfused vessel in order to maintain the vascular seal and to prevent hemorrhage. We therefore predict that in the in vivo situation (i.e. developmental angiogenesis), JBL may be more prevalent than JAIL.

The distinct temporal and spatial distribution of VE-cad, F-actin and ZO1 during JBL oscillation suggested that these proteins also partake in JBL function. In agreement with this view, interference with VE-cadherin function, as well as F-actin dynamics inhibited JBL dynamics and junction elongation.

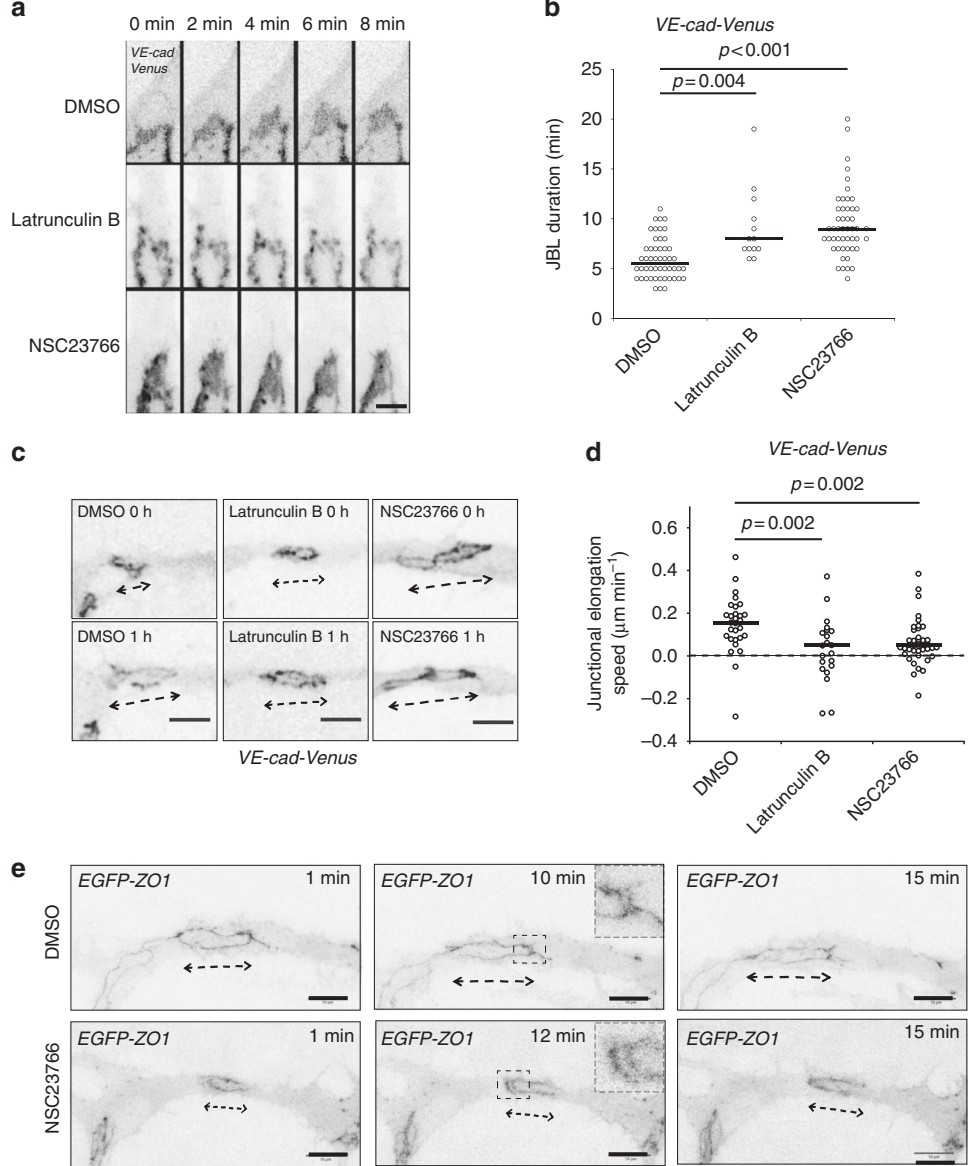

**Fig. 6** Junction elongation and JBL formation are functionally linked. **a** Still images from a movie of an VE-cad-Venus expressing embryo Tg(BAC(cdh5:cdh5-ts)) during anastomosis in the DLAV (around 32 hpf), in the presence of DMSO (1%), Latrunculin B (150 ng ml$^{-1}$) or NSC23766 (900 μM). **b** Scatter plot quantitation of the duration of the JBL. DMSO, $n = 50$ (6 movies); Latrunculin B, $n = 13$ (5 movies); NSC23766, $n = 49$ (10 movies); black lines show median values. Non-parametric Kruskal–Wallis statistical test was used. **c** Confocal images of a Tg(BAC(cdh5:cdh5-ts)) embryo during junctional elongation after DLAV anastomosis. Top panels $t = 0$ and bottom panels after 1 h incubation. **d** Quantification of the junctional elongation velocity in the presence of different chemicals using Tg(BAC(cdh5:cdh5-ts)) embryos. DMSO (1%), $n = 29$ junctions (11 embryos); Latrunculin B (150 ng ml$^{-1}$), $n = 21$ (6 embryos); NSC23766 (300 μM), $n = 41$ (11 embryos). Dotted line indicated no movement observed, black lines are medians. Non-parametric Kruskal–Wallis statistical test was used. **e** Confocal images of anastomosing DLAV of EGFP-ZO1 embryos (Tg(fli:Gal4ff$^{ubs3}$;UAS:EGFP-hZO1$^{ubs5}$)) treated with DMSO or NSC23766. Scale bar 10 μm

Previous studies have emphasized the importance for VE-cadherin in dynamic endothelial cell interactions including cell rearrangements[11,12] and cell elongation[13]. We generated a *ve-cad* mutation, which disrupts VE-cad/F-actin interaction. The phenotype observed in these mutants is slightly milder than in the null mutant suggesting that the mutant protein still allows interendothelial adhesion. Nevertheless, the increase of discontinuous junctions in SeAs illustrates an inability of mutant endothelial cells to generate multicellular tubes, which in turn suggests defects in cell rearrangements. Consistent with this view, *ve-cad$^{ubs25}$* show reduced spatial distribution of cell nuclei in the SeA. Furthermore, analysis of JBL showed that polarity and oscillatory behavior are disturbed in *ve-cad$^{ubs25}$* mutants. Taken

together, these results show that VE-cadherin actively contributes to morphogenetic cell movements via its interaction with the F-actin cytoskeleton.

The connection between VE-cad and F-actin in JBL function is also supported by F-actin interference experiments. Blocking F-actin polymerization by latrunculin-B effectively inhibits JBL formation and those JBL, which do form, show decreased oscillation. Furthermore, junctional rings do not elongate during this treatment. Inhibition of Rac1 by NSC23766 treatment did not lead to obvious defects in JBL formation, but in a reduction in their oscillation frequency as well as a defect in junction elongation. During NSC23766 treatment JBL looked relatively normal. When using a EGFP-ZO1 reporter, we also observed "double

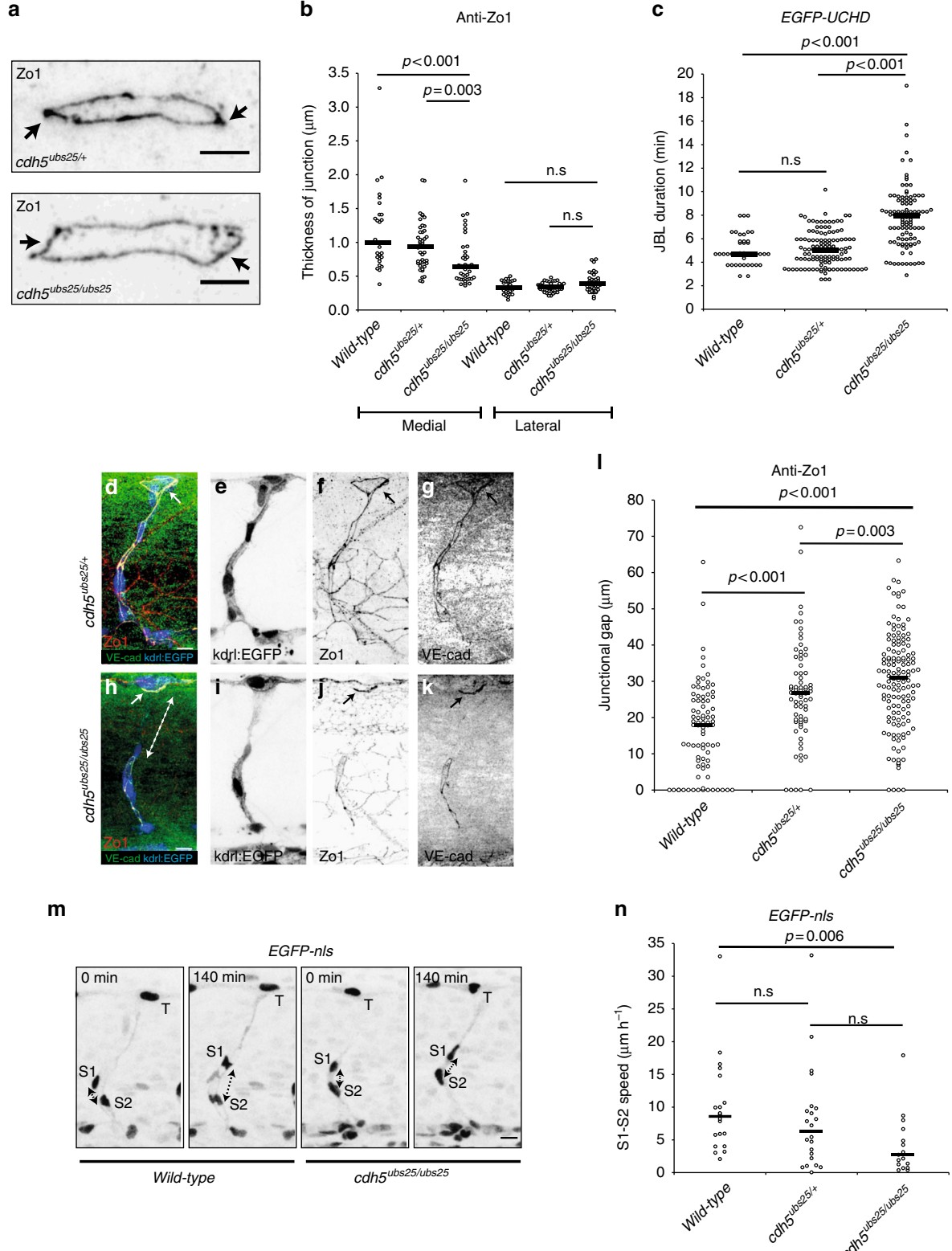

junctions", indicating that JBL are able to form a distal junction and that Rac1 may be required during later stages of JBL function.

A recent study has shown that JAIL formation requires polarized Rac1 at endothelial cell junctions in HUVECs[30]. It is therefore surprising that Rac1 inhibition does not block formation of JBL in zebrafish embryos. It is possible that even at high doses, we do not achieve full Rac1 inhibition in vivo.

Alternatively, Rac1 may have a different function in JBL than in JAIL and formation of either protrusion may require different regulators.

Taken together, we have uncovered a hitherto non-described junction-based mechanism of active cell movements, which can be used by endothelial cell (and possibly other cell types) to rearrange and adapt their shape as needed. Inhibition of JBL

**Fig. 7** Truncation of Ve-cadherin inhibits both JBL and endothelial cell remodeling. **a** Images of anti-ZO1 immunostained junctions in $cdh5^{ubs25/+}$ and $cdh5^{ubs25/ubs25}$ embryos. Arrows point to medial site of the junction. **b** Quantitation of the medial and lateral junctional thickness, based on immunostaining for ZO1; $cdh5^{ubs25/+}$, $n = 44$ junctions (17 embryos); $cdh5^{ubs25/ubs25}$, $n = 40$ (11 embryos); wild-type $n = 28$ (9 embryos). Black lines are medians. Non-parametric Kruskal–Wallis statistical test was used. **c** Quantitation of the duration of JBL based on EGFP-UCHD signal; $cdh5^{ubs25/+}$, $n = 122$ (8 embryos), $cdh5^{ubs25/ubs25}$ $n = 103$ (8 embryos) and wild-type $n = 43$ (3 embryos). All embryos carry the UAS:EGFP-UCHD transgene $Tg(fli:GFF^{ubs3};UAS:EGFP-UCHD^{ubs18})$. **d–l** $Tg(kdrl:EGFP^{s843});cdh5^{ubs25/+}$ (**d–g**) and $Tg(kdrl:EGFP^{s843});cdh5^{ubs25/ubs25}$ (**h–k**) embryos stained for VE-cadherin (rabbit antibody, green) and ZO1 (red). Individual channels are shown in inversed contrast. Both wild-type and mutant VE-cad show junctional localization (solid arrow in panels **d**, **f**, **h**, and **j**). The junctional gap in the VE-cadherin staining of a SeA in a mutant embryo ($cdh5^{ubs25/ubs25}$) is marked with dashed double arrow in panel **h**. **l** Quantification of the length of junctional gaps in control ($cdh5^{ubs25/+}$, $n = 72$ gaps, 23 embryos) and mutant ($cdh5^{ubs25/ubs25}$, $n = 139$ gaps, 33 embryos) embryos. Black lines are medians. Non-parametric Mann–Whitney statistical test was used. **m** Still images from a confocal time-lapse of endothelial nuclei ($Tg; kdrl:nlsEGFP^{ubs1}$) in wild-type or $cdh5^{ubs25/ubs25}$ embryos during SeA formation. T tip cell, S1 stalk cell 1, S2 stalk cell 2; double-headed arrow indicates the distance of S1 and S2 nuclei. **n** Quantification of movement of stalk cell 1 (S1) nuclei in relation to stalk cell 2 (S2) nuclei during cell rearrangements in SeA; $cdh5^{ubs25/+}$, $n = 22$ SeA (4 embryos); $cdh5^{ubs25/ubs25}$, $n = 17$ SeA (4 embryos); wild-type $n = 20$ SeA (4 embryos). Black lines are medians. Non-parametric Kruskal–Wallis statistical test was used. Scale bars 5 µm (**a**), 10 µm (**d**, **h**, **m**)

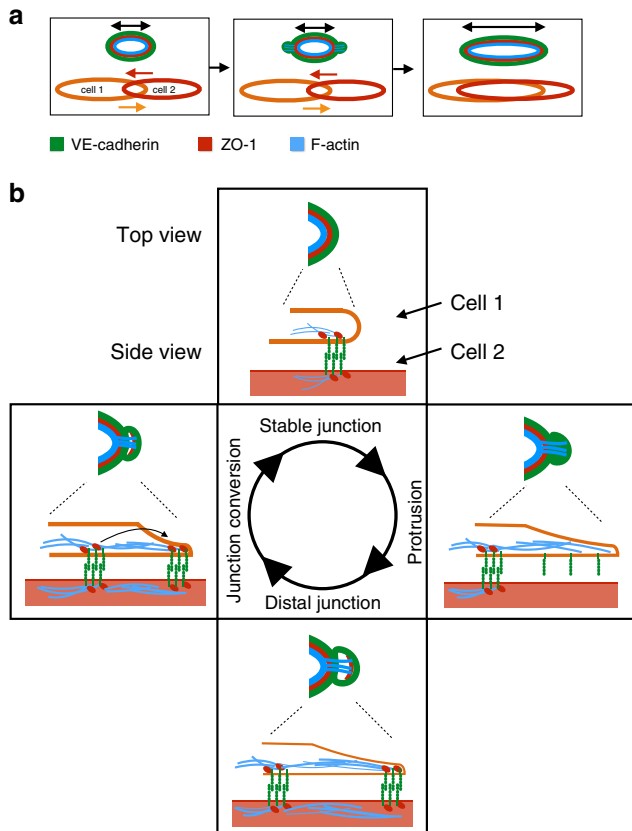

**Fig. 8** A Ratchet-like molecular mechanism of junction remodeling. **a** Stepwise elongation of an endothelial cell junction during anastomosis. As two endothelial cells move over each other (bottom), the junction becomes elongated. The three proteins investigated in this study are indicated in different colors. **b** Proposed oscillatory mechanism of JBL function. A single cycle is depicted. F-actin protrusions emanate distally from a stable junction. These protrusions also contain diffuse VE-cadherin, but not ZO1. At the distal end of the protrusion, F-actin, ZO1 and VE-cadherin are components of a newly formed junction with VE-cadherin-mediated contact to the underlying cell. Eventually, dynamic F-actin remodeling pulls the proximal junction towards the new junction

which serves as an anchor for intracellular actomyosin contractions. This cell–cell interaction may be analogous to integrin-ECM-based adhesion patches of classical lamellipodia. Furthermore, these VE-cadherin adhesion complexes give rise to a distal junction with an underlying F-actin arc. Similar F-actin arcs have been described to be essential for the function of lamellipodia during migration of human endothelial cells (HUVECs) in vitro[31]. Future studies will aim to uncover the exact mechanisms of traction force generation and transmission during JBL-driven junction elongation. While F-actin dynamics are essential for junction elongation, we have never observed prominent stress fibers in this process, suggesting that F-actin-based traction forces are acting locally rather than over the longitudinal extent of the endothelial cells.

Although we have focused our studies on JBL formation and function in the process of blood vessel anastomosis, we observed JBL also within larger caliber vessels such as the dorsal aorta at stages, when endothelial cells are extensively rearranging and undergoing cell shape changes[21]. Our studies therefore indicate that endothelial cells employ JBL as a general means for rearrangements and shape changes during blood vessel assembly and vascular remodeling. Using interendothelial adhesion for force transmission allows dynamic endothelial activities while maintaining the vascular seal. We therefore envision that JBL may underlie many morphogenetic endothelial cell behaviors during blood vessel expansion, normalization, regression and endothelial shear stress response. Remodeling and reorganization of adherens junctions is essential for developmental morphogenesis[32,33]. Whether similar JBL occur also in different tissues, besides vasculature and endothelial cells, remains to be elucidated.

## Methods

**Fish strains and maintenance**. Maintenance of fish and experimental procedures involving zebrafish embryos were carried out at the Biozentrum/Universität Basel according to Swiss national guidelines of animal experimentation (TSchV). Zebrafish lines were generated and maintained under licenses 1014H and 1014G1 issued by the Veterinäramt-Basel-Stadt. Fish strains carrying following transgenes and mutations were used in this study: $kdrl:EGFP^{s843}$ [34], $VE$-$cad$-$Venus$ ($BAC(cdh5:cdh5$-$TS)$, $fli:GFF^{ubs3}$ [19], $UAS:EGFP$-$hZO1^{ubs5}$ [19], $UAS:EGFP$-$UCHDP^{ubs18}$ [13], $UAS:mRuby2$-$UCHD^{ubs20}$ (this study), $UAS:mCLAV2$-$UCHD^{ubs27}$ (this study), $UAS:mRFP$ [35], $cdh5^{ubs25}$ (this study), and $Tg(kdrl:EGFPnls)^{ubs1}$ [36]. The fish were maintained using standard procedures and embryos obtained via natural spawning[37].

**Generation of transgenic fish lines**. The EGFP sequence of pT24xnrUAS:EGFP-UCHD was replaced by the sequence of mRuby2 (amplified from pcDNA3-mRuby2 was a gift from Michael Lin; Addgene plasmid #40260)[38] or by the sequence of mClav2 (amplified from pmClavGR2-NT; Allele Biotechnology) to generate the final plasmids pT24xnrUAS:mRuby2-UCHD and pT24xnrUAS:mClav2-UCHD respectively. These final plasmids were injected individually, together with $tol2$ RNA into $Tg(fli1:Gal4ff)^{ubs3}$ embryos and that were raised to adulthood and eventually stable transgenic fish lines $Tg(UAS:mRuby2$-$UCHD)^{ubs20}$ and $Tg(UAS:mClav2$-$UCHD)^{ubs27}$ were isolated and maintained.

function blocks these processes and results in a failure to form multi-cellular tubes and prevents formation of a patent vasculature. The salient feature of our proposed JBL model is that endothelial cells use each other as migratory substrates via VE-cadherin. Our model suggests that VE-cadherin provides an extracellular clutch, by generating an intercellular adhesion patch,

**Transient expression of mCherry-ZO1 in zebrafish embryos**. To transiently express mCherry-ZO1 in endothelial cells of zebrafish embryo, ~50 pg of plasmid *fli1ep:mCherry-ZO1*[39] was injected together with Tol2-transposase mRNA into 1–4 cell stage embryos (Tg(BAC(cdh5:cdh5-ts))). Twenty-four hours after injection healthy embryos expressing mCherry were selected, mounted in low-melting point agarose and imaged using Leica SP5 confocal microscope.

**Generation of ve-cadherin mutants**. The ve-cadherin truncation allele (*cdh5*[ubs25]) was generated using CRISPR/CAS technology[40]. We sequenced exon 12 of *cdh5* (encoding the cytoplasmic domain) of ABC, Tubingen (TU) and tupfel;long-fin (TL) strains and found a potential target sequence in ABC (5′-GGGACCTGCA CTCTATGCCATGG-3′). Target guide RNA and Cas9 protein were synthetized by standard procedures[40] and co-injected into ABC/TU embryos. Offspring of G0 fish containing germline mutations were screened by PCR analysis for the loss of a *Nco*I restriction site, which is present on the wild-type allele. For subsequent genotyping, multiplex PCR was performed using allele-specific primers:
  VE-cad-fwd: 5′-GAAACCCATATCAAACAGACCTGC-3′,
  VE-cad-rev: 5′-CAGAGCCGTCTACTCCATAAAGC-3′,
  VE-cad-ubs25-fwd: 5′-GACCTGCACTCTATGGAA-3′,
  VE-cad-wild-type-rev: 5′-GCAGGAGGTTTCTTTACC-3′.
  The genotyping protocol of *cdh5*[ubs8] has been described earlier[13]. Following primers were used:
  cdh5-FWD: 5′-TTGGTGTAACTGACAATGGGG-3′
  cdh5-REV: 5′-ACAGTCTTGGTGTTACCATTGGG-3′
  cdh5-WT-FWD: 5′-ATCCCCGTTTTCGATTCTGAC-3′
  cdh5-ubs8-REV: 5′-CTGATGGATCCAGATTGGAATC-3′

**Live imaging of zebrafish embryos**. Embryos were anesthesized using Tricaine (MS-222, 160 mg l$^{-1}$, #E10521 Sigma-Aldrich) and embedded in 0.7% low-melting point agarose (Sigma-Aldrich) supplemented with Tricaine in glass-bottom dish. After the agarose solidified, it was overlaid with E3-medium supplemented with Tricaine. All the imaging was performed at 28.5 °C. The imaging was performed using Leica SP5 Matrix confocal microscope equipped with resonance scanner using ×63 NA 1.2 or ×40 NA 1.1 water immersion objectives. For imaging of JBL, the time points were 60–120 s intervals and in case of double-color imaging, 12–60 s.
  For the pharmacological experiments, the treatment of embryos with inhibitors (DMSO 1%, Latrunculin B (150 ng ml$^{-1}$), NSC23766 (300–900 μM)) begun 1 h prior to embedding into low-melting point-agarose and confocal imaging. The inhibitors were present throughout the whole experiment.

**Generation of polyclonal rat anti-zf-VE-CAD antibodies**. A cDNA fragment encoding a polypeptide comprising the extracellular domain of zebrafish VE-cad (Ala22 to Lys464) was expressed in *E. coli* using the T7 expression system. The protein was purified on Ni-charged IMAC resin (BioRad) under denaturing conditions. Antiserum was raised in rats by ThermoFisher Scientific using standard immunization procedures.

**Immunofluorescence analysis**. Embryos were fixed with 2% paraformaldehyde (Electron Microscopy Sciences) in PBST (PBS + 0.1% Tween-20) at room temperature for 90 min, and immunostained using following protocol: Fixation with 2% PFA/PBST (PBS + 0.1% Tween-20) for 90 min at room temperature followed by washes with PBST. After permeabilization (PBST + 0.5% Triton X-100, 30 min), the samples were blocked (PBST + 0.1% Triton X100 + 10% normal goat serum + 1%BSA + 0.01% Sodium Azide, overnight, 4 °C). Subsequently, primary antibodies were added (diluted in Pierce Immunostain enhancer, #46644, Thermofisher Scientific) and incubated overnight at 4 °C. After several washes with PBST at room temperature, the secondary antibodies were added (1:2000 dilution in Pierce staining Enhancer) and incubated overnight at 4 °C. After several washes with PBST at room temperature, the embryos were mounted onto glass-bottom dishes using low-melting point agarose.
  Mouse anti-hZO1 (dilution 1:400, Invitrogen #33–9100; use in zebrafish in ref. [36], rat anti-VE-cad (dilution 1:500) and rabbit anti-VE-cad (dilution 1:500) primary antibodies were used. Rat anti-VE-cad was validated by immunofluorescence by using *ve-cad* null mutant (*cdh5*[ubs8], Tg(kdlr:EGFP[ps843])) embryos[13] as control for specificity (Supplementary Fig. 2). Rabbit anti-VE-cad has been described and validated previously[36] and also validated with *ve-cad* null mutants[13]. Fluorescent secondary antibodies Alexa-568 goat anti-mouse IgG, Alexa-633 goat anti-rat IgG, and Alexa-633 goat anti-rabbit IgG (all from Invitrogen) were used.

**Photoconversion experiment**. Twenty-four hours post fertilization zebrafish embryos (*Tg(Fli:GFF*[ubs3];*UAS:mClav2-UCHD*[ubs27])) were embedded in 0.7% low-melting point agarose onto 35 mm glass bottom dishes. Tip cells of vascular sprouts of segmental arteries were photoconverted on a Leica SP5 confocal microscope using a ×40 NA 1.1 water immersion objective. Photoconversion was performed with a 405 nm laser (20% power) until no further increase in converted UCHD-mClav2 signal was observed (conversion time 10–30 s). After this the embryos were allowed to develop for ~4 h before imaging of anastomosis events in DLAV.

**Junction elongation experiment**. Junctional elongation was analyzed by observing anastomosis and elongation of isolated junctional rings during DLAV formation. Inhibitor treatments Latrunculin B (150 ng ml$^{-1}$), NSC23766 (300–900 μM) or DMSO (1%) were applied 1 h before mounting of embryos into 0.7% low-melting point agarose and imaging the junctions for 1–2 h on a Leica SP5 (×40 NA 1.1 water immersion objective).

**Cell rearrangement experiment**. Embryos carrying nuclear GFP (Tg(kdrl: EGFPnls[ubs1])) were used. Confocal Z-stacks were obtained as described above in 12–14 min intervals. To control small variations in the developmental phases of the individual SeA, $t = 0$ min was the time point when tip cell had reached the dorsal side and started to sprout in anteroposterior directions. The endpoint for analysis was 10 time points later ($t = 10$; 120–140 min). At $t = 0$ and $t = 10$ the distance ($d$) of stalk cell S1 and S2 was measured using FIJI. Speed of movement of S1 and S2 in relative to each other was calculated using equation:

$$\text{speed} = \frac{|d_{t0} - d_{t10}|}{\Delta t} \quad (1)$$

**Image analysis and preparation**. Image analysis and measurements were performed using FIJI. Deconvolution was performed using Huygens Remote Manager software[41]. Maximum Z-projections were used. Noise was reduced using Gaussian filtering (radius 1.0) and background subtracted (rolling ball radius 50) using FIJI. Contrast and brightness of images were linearly adjusted. Kymographs were generated from the sum Z-projections of time-lapse series using FIJI. Perpendicular straight line across the junction was drawn and kymograph generated using reslice tool. Colocalization analysis was performed using FIJI. First, background was subtracted (rolling ball radius 25), regions of interests (ROIs) of separate cell–cell junctions were selected and then colocalization analysis was done on these ROIs using Colocalization Test plugin with Fay image randomization ((written by Tony Collins, McMaster University, Hamilton, Canada). Publication figures were prepared using FIJI, OMERO Fig. and Adobe Illustrator.

**Statistical analyses**. Statistical analyses were performed using Microsoft Excel, IBM SPSS statistics 22 and GraphPad Prism version 6.05 for Windows software. Non-parametric two-sided Kruskal–Wallis *H*-test, Mann–Whitney *U*-test and binomial probability (photoconversion experiments, test probability 0.5) were used. The data reasonably met the assumptions of the tests. In Fig. 1e, f, where the data was non-normal and heteroscedastic similar *p*-values were obtained with Kruskal–Wallis *H*-test, Welch's *t*-test and median test.
  No statistical power analysis was used to determine samples size. Systematic randomization was not used. Experiments with cdh5[ubs25] (Fig. 7) were performed essentially blinded as the genotype was determined after data capture and analysis. In all other experiments blinding was not used. Samples of low technical quality were excluded from the subsequent analyses. In all figures (exception Fig. 2f and Supplementary Fig. 3d) the individual data points are plotted and median indicated with horizontal line as has been recommended[42]. In Fig. 2f, Supplementary Fig. 1a and c, the data are binned and the number of events in the given bin is plotted.
  To formally analyze periodicity of F-actin intensity fluctuations, we calculated the autocorrelation function (ACF) using IBM SPSS statistics 22 software. Next, we analyzed the level of noise in the fluctuations (phase diffusion) that gradually reduces the oscillation of the ACF. To quantify this effect, we fitted each of the ACFs using a sinusoidal function enveloped by an exponential decay using Igor Pro 6.37 (WaveMetrics Inc, Lake Oswego, OR, USA). The function used was:

$$\text{ACF}(t) = A \times \sin(f \times t + \theta) \times e^{\left(-\frac{t}{\tau}\right)} \quad (2)$$

where *t* is the lagtime of the ACF, *A* is amplitude, *f* the frequency, *θ* is the phase, and *τ* is the characteristic lifetime of the decay.

## Data availability

The data that support the findings of this study are available from the corresponding authors upon reasonable request.

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

## Acknowledgements

We thank Kumuthini Kulendra for fish care and the Imaging Core Facility of the Bio-zentrum (University of Basel) for microscopy support. We thank Johanna Ivaska for support and acknowledge Zebrafish Core Facility (Turku Centre for Biotechnology, University of Turku and Åbo Akademi University). This work has been supported by the Kantons Basel-Stadt and Basel-Land and by a grant from the Swiss National Science Foundation to M.A. I.P. was supported by a post-doctoral fellowship from the Finnish Cultural Foundation and Foundations' Post-Doc Pool. M.L., C.W., and L.S. were supported by a Fellowship of Excellence, Biozentrum, University of Basel.

## Author contributions

H.G.B., I.P., and M.A. conceived the idea and directed the work. I.P. and H.G.B. designed the experiments. I.P., L.S., M.L., C.W., and D.H. performed experiments. D.B., A.K.L., and B.M.H. provided unpublished reagents. C.G. helped with data analysis. I.P. and H.G.B. wrote the manuscript. All authors reviewed the manuscript.

## Additional information

**Competing interests:** The authors declare no competing interests.

