## [Peer Review File · Nature Communications]

Reviewers' comments:

Reviewer #1 (Remarks to the Author):

The authors have analysed the mechanisms of cell rearrangements during multicellular blood vessels formation in zebrafish. They propose a ratchet-like mechanism supported by VE-Cad and F-Actin dynamics promoting junction elongation. This proposition is in part supported by elegant in vivo high-resolution time-lapse microscopy and by the inhibition of Rac, VE-Cad activities and F-Actin formation. While the authors do not provide extensive mechanistic insights on the molecular mechanisms supporting the F-Actin and VE-Cad dynamics, this work would be a strong candidate for Nature Communications once the following comments are experimentally addressed.

Major comments

1. Rigorous quantifications of the photoconverted and non-photoconverted F-Actin levels are required to support the following 2 statements: "The JBL originated as a F-actin protrusion at the front end of the "donor" cell (Fig. 4D, red channel). This was associated with accumulation of F-actin to the site within the (non photo-converted) recipient cell (Fig. 4D, green channel) (this occurred in 9 out of 10 analyzed events).". "These results show that cell interactions at JBL occur in a two-step process: an initial actin F-accumulation takes place at the front end of the elongating junction, which subsequently induces a secondary F-actin accumulation in the underlying cell." It would also be helpful if the authors would describe what is observed in the absence of JBL. In other words, is the interplay between the two cell F-actin pools at their junction specific to JBL? Furthermore, can the authors demonstrate that they only photoconvert the JBL? It is rather surprising that they can photoconvert such a restricted apical-basal domain region without photo-convert the neighbouring cell in contact more basally.

2. The following is not supported by the data presented: "In contrast, EGFP-ZO1 showed a more defined localization and initially remained associated with the junction at the proximal end of the protrusion (Fig. 5D 0 to 36 sec.)." The authors do not show the 36s time-point for VE-Cad. Furthermore, the colocalisation analysis between ZO-1 and VE-Cad is only shown on a still image (Fig 5). A time-lapse analysis is needed to further support the authors' claim.

3. How are JBL defined upon Rac inhibition? It is very difficult to see any lamellipodia-like structures in Figure 6A. Time-lapse movies need to be provided. How are VE-Cad and ZO-1 localised upon F-Actin and Rac inhibition treatments? Could the authors describe the local and temporal dynamics of F-Actin (See Fig 3) upon Rac inhibition? If experimentally possible, the authors should determine whether Rac is required in the donor or the recipient cells.

4. The following statement is not supported by the authors' results (page 10): "and that the VE-cad/F-actin interaction is essential for JBL function and junction elongation." The authors at least need to show that the mutant VE-Cad protein is correctly localised. Is the VE-Cad

homozygote mutant JBL duration significantly different from the wild-type one? The authors need to show and to compare the JBL duration in VE-Cad mutant and upon Rac inhibition in the same graph. Are there less JBL in the VE-Cad mutant vessels? How is junction elongation speed affected in VE-Cad mutant vessels? Overall a more extensive comparison of the Rac inhibition and VE-Cad mutant conditions is necessary to support their action in a common process.

5. The authors do not explicit why the longer duration of JBL in VE-Cad mutant blood vessels support a ratchet-like mechanism. Would one not expect that in the absence of a clutch the duration would be much shorter? The description of the LatA, Rac inhibition and VE-Cad mutant conditions should be schematised in Figure 7.

Minor comments:

1. A schematic of the unicellular versus multicellular tubes would be helpful in Figure 1 for the readers.
2. "However, movement of the junctions was also seen in perfused vessels.". please add "Figure Sup 1C".
3. Time-stamps and arrows pointing at the structures of interest need to be added on movies.

Reviewer #3 (Remarks to the Author):

In this paper, Paatero and colleagues study how endothelial cells re-arrange and move in vivo using the Zebrafish as a model system. The authors investigate the role of endothelial cell junctions and the actin cytoskeleton in this process and use advanced live cell imaging and photo-conversion experiments to demonstrate their points.

The topic of the paper: how endothelial cell migrate and rearrange during vascular development, is highly relevant to the angiogenesis research field, and the cellular and molecular mechanisms of endothelial cell migration are still poorly understood and rarely studied in this detail in vivo. The authors observe that endothelial cell movements are associated with oscillating lamellipodia-like structures that emerge at endothelial cell junctions (referred to by the authors as junctional based lamellopodia or JBL). These structures are primarily oriented in the direction of migration and are formed by actin-based protrusions that also contain VE-Cadherin. The authors propose a model in which JBL activity is required for VE-Cadherin deposition at the front of migrating cells. Newly deposited VE-Cadherin, in turn, is thought to provide anchoring point for the migrating cell and is required for the formation of new EC junctions. As the endothelial cell pulls itself forward, the old, proximal, junction is moved towards the newly established junction.

Similar models involving actin based lamellopodial protrusions with cadherin-rich membranes have previously been described in vitro in HUVECs (Taha, A; Mol Biol Cell 2014

and Hayer, A; Nat cell Biol 2016), however, this report describes such structures - to my knowledge for the first time - in vivo, which is a major advance in the field. The authors provide convincing evidence for the existence of junctional-based lamellopodia formation and associated VE-Cadherin depositions during endothelial cell migration/re-arrangements in vivo, but fail to demonstrate the in vivo relevance of their observation - which is the only major concern I have. If junctional-based lamellopodia formation and associated VE-Cadherin depositions are really crucial for endothelial cell migration and re-arrangements in vivo, as the authors propose, one would expect that interference with the formation of those structures would result in correspondingly severe vascular phenotypes - at least one would expect a major delay of vascular development. However, despite the fact that inhibition of F-actin polymerization led to defects in JBL formation and inhibition of Rac1 activity in JBL dynamics, the authors fail to describe corresponding in vivo-phenotypes in the present manuscript and only describe defects in "junctional elongation". I am well aware that it might be challenging to find experimental conditions that effectively block JBL formation or associated VE-Cadherin deposition, and, as a consequence, cause vascular abnormality in vivo (that are not secondary), but I think the authors miss an important chance to demonstrate the in vivo relevance of their model. The ve-cadubs25 mutant, which prevents VE-Cadherin-F-actin interactions, and in which F-actin protrusions oscillate slower would, for example, be expected to produce such an experimental condition. The authors state that in homozygosity, the ve-cadubs25 phenotype is similar to that of ve-cadherin null mutants, but they show rather weak evidence for this in S3 and refer to "data not shown". In my opinion, even the VE-Cadherin null phenotype (Sauter et al. Cell Rep 2014) is "surprisingly mild" for a component thought to be strictly required for endothelial cell migration.

The paper is clearly written and well structured, and the figures illustrate the authors' points well for the most part. The experiments are technically sound and the authors use state-of-the-art live imaging approaches to reach their conclusions.

I am further wondering if the proposed mechanism is conserved and of relevance for endothelial cell migration in other species? Endothelial specific deletion of Rac1 in postnatal mice, for example, results in a very mild retinal phenotype that is not compatible with a major role in endothelial cell migration (Nohata et al. Dev Biol 2016), yet Paatero and colleagues find that pharmacological Rac1 inhibition in ZF interferes with JBL dynamics and junctional elongation. Some comments on this would be welcome.

In conclusion, I think that the paper addresses a highly relevant topic and provides a plausible model for endothelial cell migration and rearrangements, but it lacks convincing evidence for a corresponding vascular importance (phenotype) in vivo. If the proposed mechanism is relevant, inhibition of the underlying cellular processes would be expected to result in a global endothelial cell migration phenotype. If such a phenotype is observed by the authors, it should be clearly presented. Alternatively, redundancy and compensatory mechanism might explain the lack of a global endothelial cell migration phenotype, but they are not sufficiently discussed.

Mino points:

Typo: Supple Figure 1C, the Y-axis legend says: ...inflated... should probably be inflated I assume?

Typo: 318 prominent stress fibers during in this process

Rebuttal to reviewer comments

Reviewers' comments in plain font, author reply in **bold/green**

We thank the reviewers for their overall positive responses and greatly appreciate their constructive criticism, which has helped us to improve our study. Below, please find our point-by-point responses to the reviewer's comments.

Reviewer #1 (Remarks to the Author):

The authors have analysed the mechanisms of cell rearrangements during multicellular blood vessels formation in zebrafish. They propose a ratchet-like mechanism supported by VE-Cad and F-Actin dynamics promoting junction elongation. This proposition is in part supported by elegant in vivo high-resolution time-lapse microscopy and by the inhibition of Rac, VE-Cad activities and F-Actin formation. While the authors do not provide extensive mechanistic insights on the molecular mechanisms supporting the F-Actin and VE-Cad dynamics, this work would be a strong candidate for Nature Communications once the following comments are experimentally addressed.

Major comments

1. Rigorous quantifications of the photoconverted and non-photoconverted F-Actin levels are required to support the following 2 statements: “The JBL originated as a F-actin protrusion at the front end of the “donor” cell (Fig. 4D, red channel). This was associated with accumulation of F-actin to the site within the (non photo-converted) recipient cell (Fig. 4D, green channel) (this occurred in 9 out of 10 analyzed events).”. “These results show that cell interactions at JBL occur in a two-step process: an initial actin F-accumulation takes place at the front end of the elongating junction, which subsequently induces a secondary F-actin accumulation in the underlying cell.” It would also be helpful if the authors would describe what is observed in the absence of JBL. In other words, is the interplay between the two cell F-actin pools at their junction specific to JBL? Furthermore, can the authors demonstrate that they only photoconvert the JBL? It is rather surprising that they can photoconvert such a restricted apical-basal domain region without photoconvert the neighbouring cell in contact more basally.

Q1.1: Rigorous quantifications of the photoconverted and non-photoconverted F-Actin levels are required to support the following 2 statements: “The JBL originated as a F-actin protrusion at the front end of the “donor” cell (Fig. 4D, red channel). This was associated with accumulation of F-actin to the site within the (non photo-converted) recipient cell (Fig. 4D, green channel) (this occurred in 9 out of 10 analyzed events).”.

We agree this reviewer's concern about the conclusion drawn from Fig. 4D. While Figure 4B and 4C clearly show that the F-actin protrusion of the JBL form at the front of the “donor” cell, the subsequent accumulation of F-actin in the underlying cell is much less convincing. Although we have tried to quantify differences in fluorescent intensities, these attempts failed due to too small dynamic differences in fluorescence and the limited

temporal resolution. We have therefore removed the above statement from the text and panel D from Figure 4.

The reviewer raises an important point when asking about F-actin in the absence of JBL. Here, we observe a tight colocalization of junctional F-actin from the converted and non-converted F-actin pools. This is indicated now by an asterisk in Figure 4B and in the text. Therefore, it appears that JBL are unique in that they show a mismatch between junctional F-actin pools between two endothelial cells.

The reviewer is wondering how we can differentiate between two contacting endothelial cells during photoconversion. To achieve controlled photoconversion of a “single cell”, we take advantage of the physical separation of the angiogenic sprouts during the early sprouting phase (at 24-26 hpf). We photoconvert the entire endothelial tip cell (and possibly to some extent a neighboring stalk cell) of a single sprout. 2 to 4 hours later (during anastomosis) the photoconverted cell will engage into contact with a non-converted tip cell from the neighboring segment. This allows us to distinguish between the F-actin pools of the two tip cells.

2. The following is not supported by the data presented: “In contrast, EGFP-ZO1 showed a more defined localization and initially remained associated with the junction at the proximal end of the protrusion (Fig. 5D 0 to 36 sec.)” The authors do not show the 36s time-point for VE-Cad. Furthermore, the colocalisation analysis between ZO-1 and VE-Cad is only shown on a still image (Fig 5). A time-lapse analysis is needed to further support the authors’ claim.

Due to smaller spectral separation of Venus and mRuby (VE-cad vs. F-actin) compared to GFP and mRuby (ZO-1 vs. F-actin), we had to sample VE-Cad movies with somewhat slower time resolution than ZO-1 movies. We have now provided time-lapse analysis of direct comparison of mCherry-ZO-1 and VE-cad-Venus (new Figure 5F). This analysis confirms our previous observations.

3. How are JBL defined upon Rac inhibition? It is very difficult to see any lamellipodia-like structures in Figure 6A. Time-lapse movies need to be provided. How are VE-Cad and ZO-1 localised upon F-Actin and Rac inhibition treatments? Could the authors describe the local and temporal dynamics of F-Actin (See Fig 3) upon Rac inhibition? If experimentally possible, the authors should determine whether Rac is required in the donor or the recipient cells.

Time-lapse movies for Figure 6A are now provided as supplemental movies.

JBL was defined as a local thickening of VE-Cad at the cell junction. We acknowledge that the JBL morphology in inhibitor treated embryos is not normal. The defects in local junctional dynamics were further confirmed in the analyses of F-actin oscillations. We analysed effects of NSC23766 on local actin dynamics by using kymographs. The dynamics of UCHD-GFP were clearly altered by NSC and LatB treatments (Supplemental Fig.3). We have now also performed advanced analysis utilizing autocorrelation of UCDH-GFP intensity oscillations, and fitting the resulting autocorrelation functions into a theoretical model (a sinusoidal function enveloped by an exponential decay). The function used was:

$$ACF(t) = A * \sin(f * t + \theta) * e^{\left(-\frac{t}{\tau}\right)}$$

The decay of the ACF (tau parameter of the function) was different, indicating that in the NSC and LatB treated embryos the junctional F-actin oscillations were more irregular.

As proposed by the reviewer we have performed *in vivo* time-lapse imaging of Rac1 inhibition using the EGFP-ZO-1 reporter. These experiments confirm that acute NSC23766 treatment inhibits junction elongation during anastomosis. Nevertheless, we find that ZO-1 can still form dual junctions, representing the proximal (old) and distal (new) junctions. These observations suggest that Rac1 is not required for formation of JBL and the formation of distal junctions.

As the reviewer points out, it would be important to differentiate between Rac1 function in donor and host cells. We don't think that this type of analysis can be done by acute inhibition of Rac1 *in vivo*. We are in the process of developing new transgenic tools which will allow us to interfere with Rac1 in a more controlled fashion. However, we feel that these future experiments are outside the scope of our present work.

4. The following statement is not supported by the authors' results (page 10): "and that the VE-cad/F-actin interaction is essential for JBL function and junction elongation." The authors at least need to show that the mutant VE-Cad protein is correctly localised. Is the VE-Cad homozygote mutant JBL duration significantly different from the wild-type one? The authors need to show and to compare the JBL duration in VE-Cad mutant and upon Rac inhibition in the same graph. Are there less JBL in the VE-Cad mutant vessels? How is junction elongation speed affected in VE-Cad mutant vessels? Overall a more extensive comparison of the Rac inhibition and VE-Cad mutant conditions is necessary to support their action in a common process.

Q4.1: The authors at least need to show that the mutant VE-Cad protein is correctly localised.

We have included immunofluorescence analyses, which indicate, that the truncated VE-cad protein is correctly localized to cell-cell junctions (shown in Figure 7H and K), indicated with an arrow).

Q4.2: Is the VE-Cad homozygote mutant JBL duration significantly different from the wild-type one?

In the previous data set we focused our comparison on *ubs25*^{+/-} and *ubs25*^{-/-} embryos, and used wild-type samples only as a reference. Now, we analysed more data on WT samples to enable direct comparison, and as expected, longer duration of JBLs in VE-cad homozygote mutants was observed (Fig7C).

Q4.3: The authors need to show and to compare the JBL duration in VE-Cad mutant and upon Rac inhibition in the same graph.

The direct comparison of Ve-cad mutants and Rac inhibitor is not possible due to different experimental conditions and control groups. As a key issue, we have used different transgenic lines; although VE-cad-Venus transgene is superior to UCHD-GFP in

unambiguously defining cell-cell junctions during anastomosis, it would have rescued any defects caused by *ubs25* mutation. The potential of the VE-cad-Venus transgene to rescue VE-cad mutants has recently been published (Lagendijk et al., *Nat. Comm.* (2017)).

Q4.4: Are there less JBL in the VE-Cad mutant vessels?

Measuring the occurrence of JBL would be extremely interesting. However, at this point counting and calculating JBL is complicated by several issues. For one, JBL are transient and we have to image them at such high resolution that we only cover portions of junctional rings, which in turn makes it difficult to normalize the occurrence to the length of the cell junction in an *in vivo* setting. In our study, we have therefore resorted to measuring the thickness of junctions along their circumference. We will follow up on the effect of loss of VE-cad in JBL function in a future study.

Q4.5: How is junction elongation speed affected in VE-Cad mutant vessels?

Defects in junction elongation lead to the increased occurrence and length of interjunctional gaps during ISV formation (Sauteur et al., *Cell Rep* (2014)). Therefore, we have used these gaps as a downstream readout for the speed of junctional elongation in *ve-cad^{ubs25}* mutants - displayed in Figure 7L. Indeed, the *ve-cad^{ubs25}* mutants have larger interjunctional gaps, indicating slower elongation of the junctions.

5. The authors do not explicit why the longer duration of JBL in VE-Cad mutant blood vessels support a ratchet-like mechanism. Would one not expect that in the absence of a clutch the duration would be much shorter? The description of the LatA, Rac inhibition and VE-Cad mutant conditions should be schematised in Figure 7.

-- The fact that we see a reduced frequency of oscillation in the *ve-cad* mutant conditions is not in conflict with a ratchet mechanism. At this point we can only speculate about the regulation of JBL oscillations. In the simplest model of oscillation, the oscillations are generated by delayed negative-feedback signal, aka. time-delayed negative-feedback oscillator (Novák, B. & Tyson, J. J. *Design principles of biochemical oscillators*. *Nat. Rev. Mol. Cell Biol.* 9, 981–91 (2008)). Following this theoretical framework, the oscillating behaviour of junctions would imply that there is delayed negative signal that controls the duration of the oscillations. Our data from analysis of F-actin dynamics of the remodelling junctions indicate defects in the F-actin oscillations after inhibitor treatments. One possible explanation for increased duration of JBLs would be that there is a specific local, and transient, negative feedback signal that inhibits the growth phase of JBLs and results in retraction phase, and the induction of this signal would be diminished in the mutant and inhibitor treated embryos.

Minor comments:

1. A schematic of the unicellular versus multicellular tubes would be helpful in Figure 1 for the readers.

We have included this now in Figure 1.

2. “However, movement of the junctions was also seen in perfused vessels.”. please add “Figure Sup 1C”.

This has been corrected.

3. Time-stamps and arrows pointing at the structures of interest need to be added on movies.

These have now been included in the supplementary movies.

Reviewer #3 (Remarks to the Author):

In this paper, Paatero and colleagues study how endothelial cells re-arrange and move in vivo using the Zebrafish as a model system. The authors investigate the role of endothelial cell junctions and the actin cytoskeleton in this process and use advanced live cell imaging and photo-conversion experiments to demonstrate their points.

The topic of the paper: how endothelial cells migrate and rearrange during vascular development, is highly relevant to the angiogenesis research field, and the cellular and molecular mechanisms of endothelial cell migration are still poorly understood and rarely studied in this detail in vivo. The authors observe that endothelial cell movements are associated with oscillating lamellipodia-like structures that emerge at endothelial cell junctions (referred to by the authors as junctional based lamellopodia or JBL). These structures are primarily oriented in the direction of migration and are formed by actin-based protrusions that also contain VE-Cadherin. The authors propose a model in which JBL activity is required for VE-Cadherin deposition at the front of migrating cells. Newly deposited VE-Cadherin, in turn, is thought to provide anchoring point for the migrating cell and is required for the formation of new EC junctions. As the endothelial cell pulls itself forward, the old, proximal, junction is moved towards the newly established junction.

Similar models involving actin based lamellopodial protrusions with cadherin-rich membranes have previously been described in vitro in HUVECs (Taha, A; Mol Biol Cell 2014 and Hayer, A; Nat cell Biol 2016), however, this report describes such structures - to my knowledge for the first time - in vivo, which is a major advance in the field.

The salient features of JBL are their oscillatory behaviour, their emergence from intact EC junctions, their orientation along the blood vessel axis and their promotion of cell/junction elongation. As the reviewer points out, lamellipodial protrusions of endothelial cells, which are associated with EC junctions termed JAILs and “cadherin fingers”, respectively, have been previously described in HUVEC cells. We have noticed a recent publication by the group of Hans Schnittler (Cao et al., *Nat. Comm.* (2017)), which further describes the role of so-called JAILs in endothelial cell behavior. By studying junctional dynamics in cell culture

experiments and histological analysis of fixed embryonic murine blood vessels, the authors reach conclusions similar to ours.

Despite their similarities, JBL are structurally and functionally different from JAIL. JAIL have been originally described *in vitro* – e.g. in HUVECs und subconfluent conditions. Thus, the term JAIL (“junction associated intermittent lamellipodia”) has been coined to describe a local breakdown of junctions followed by F-actin protrusions and de novo formation of a distal junction (Abu Taha et al., *MBC* (2014)). We have never observed such junctional breakdown in zebrafish *in vivo* and rather suggest that JAILs occur under subconfluent conditions, when junctional stability is generally decreased.

To emphasize these JBL characteristics and their distinction from similar protrusion observed *in vitro* (i.e. JAIL), we have included a new paragraph in the discussion.

The authors provide convincing evidence for the existence of junctional-based lamellopodia formation and associated VE-Cadherin depositions during endothelial cell migration/re-arrangements *in vivo*, but fail to demonstrate the *in vivo* relevance of their observation - which is the only major concern I have. If junctional-based lamellopodia formation and associated VE-Cadherin depositions are really crucial for endothelial cell migration and re-arrangements *in vivo*, as the authors propose, one would expect that interference with the formation of those structures would result in correspondingly severe vascular phenotypes - at least one would expect a major delay of vascular development. However, despite the fact that inhibition of F-actin polymerization led to defects in JBL formation and inhibition of Rac1 activity in JBL dynamics, the authors fail to describe corresponding *in vivo*-phenotypes in the present manuscript and only describe defects in “junctional elongation”. I am well aware that it might be challenging to find experimental conditions that effectively block JBL formation or associated VE-Cadherin deposition, and, as a consequence, cause vascular abnormality *in vivo* (that are not secondary), but I think the authors miss an important chance to demonstrate the *in vivo* relevance of their model. The *ve-cadubs25* mutant, which prevents VE-Cadherin-F-actin interactions, and in which F-actin protrusions oscillate slower would, for example, be expected to produce such an experimental condition. The authors state that in homozygosity, the *ve-cadubs25* phenotype is similar to that of *ve-cadherin* null mutants, but they show rather weak evidence for this in S3 and refer to “data not shown”.

We agree with the reviewer, that in our first submission we had focused on the cell biological side of *ve-cad^{ubs25}* mutant and the overall vascular phenotype was somewhat superficially described. We have now provided more in-depth phenotypic description of the VE-cad truncation mutant phenotype (Fig. 7 and supplemental Figure 5).

In short, compared to the homozygous *ve-cad^{ubs8}* (=null) mutant the *ve-cad^{ubs25}* allele shows a hypomorphic phenotype. While in null mutants the disorganization of the endothelial architecture results in a complete abrogation of circulation in the DA and PCV (Sauteur et al., (2014), Sauteur et al., (2017)), the *ve-cad^{ubs25}* mutants display limited blood flow in the axial vessels as well as in the ISVs. Consistent with these observations, injection of a “tailless” VE-cad-GFP fusion somewhat ameliorates the null mutant phenotype (Sauteur et al., (2014); L.S and H.G.B., unpublished observations). Nevertheless, both alleles affect junctional remodelling and thus inhibit multicellular tube formation during angiogenesis, ultimately resulting in a dysfunctional vasculature and complete lethality (Fig. 7M and N).

In my opinion, even the VE-Cadherin null phenotype (Sauteur et al. Cell Rep 2014) is “surprisingly mild” for a component thought to be strictly required for endothelial cell migration.

The notion that the ve-cad phenotype is mild may be due to anatomical differences in fish and mouse – in particular when considering the mouse retina, which provides a wide open “playing field” for angiogenic sprouting. In the zebrafish embryo space is limited and the endothelial cells have to squeeze between somites, notochord etc.. This spatial constraint forces the cells into a seemingly normal vascular pattern suggestive of a mild phenotype. However, this appearance is misleading and at higher resolution it becomes clear that the *ubs8* and *ubs25* mutant phenotypes are quite consistent with observations made in mice. E.g. Carmeliet et al., (*Cell*, 1999) concluded that lack of VE-cad leads primarily to defects in vascular remodelling and maturation. At higher resolution, Gaengel et al., (*Dev.Cell* 2012) noticed that tip cells separated from stalk cells at the angiogenic front in post-natal mouse retinas (e.g. Figure 4). This separation is also observed in zebrafish null mutants (Sauteur et al., 2014) as well as in *ubs25*.

Nevertheless, the “mild” phenotype also surprised and prompted us to look for other adhesion molecules, which may act redundant with VE-cad or can compensate for its loss. By comparing loss of function alleles of *ve-cad* and *esama* in different combinations, we found that both proteins act redundantly during vascular tube formation and anastomosis (Sauteur et al., (*Development*, 2017). Double mutants of these genes leads to a massive disorganisation of the endothelial architecture as well as a complete failure of filopodia-mediated contact formation between tip cells.

Moreover, we observed a two-fold increase of *esama* mRNA levels in *ve-cad* null mutants, suggesting that elevated *Esama* may partially compensate for the lack of VE-cad protein.

The paper is clearly written and well structured, and the figures illustrate the authors' points well for the most part. The experiments are technically sound and the authors use state-of-the-art live imaging approaches to reach their conclusions.

We appreciate these encouraging comments!

I am further wondering if the proposed mechanism is conserved and of relevance for endothelial cell migration in other species? Endothelial specific deletion of *Rac1* in postnatal mice, for example, results in a very mild retinal phenotype that is not compatible with a major role in endothelial cell migration (Nohata et al. Dev Biol 2016), yet Paatero and colleagues find that pharmacological *Rac1* inhibition in ZF interferes with JBL dynamics and junctional elongation. Some comments on this would be welcome.

We do not propose that VE-cad is important for regular endothelial cell migration as – for example – in tip cell migration, which is unaffected in *ve-cad* mutants. We rather focus on its role on endothelial cell movements, which are based on junctional remodelling such as cell rearrangements and cell shape changes.

The mild defects in the retinal vasculature caused by postnatal endothelial deletion of *Rac1* are indeed surprising, given the fact that *Rac1* is essential for numerous F-actin based cell dynamics. Nevertheless, Nohata et al. describe several vascular defects upon late embryonic as well as postnatal *Rac1* excision, which are consistent with defects in vascular remodelling and cell rearrangements, including branching defects and hemorrhages.

While we understand the reviewer's interest in this topic, we prefer not to discuss this in detail, because – at this point – it would appear too speculative and distract from the key findings of our studies.

In conclusion, I think that the paper addresses a highly relevant topic and provides a plausible model for endothelial cell migration and rearrangements, but it lacks convincing evidence for a corresponding vascular importance (phenotype) in vivo. If the proposed mechanism is relevant, inhibition of the underlying cellular processes would be expected to result in a global endothelial cell migration phenotype. If such a phenotype is observed by the authors, it should be clearly presented. Alternatively, redundancy and compensatory mechanism might explain the lack of a global endothelial cell migration phenotype, but they are not sufficiently discussed.

As discussed above, we expanded our data as well as the discussion with respect to the relevance of JBL in overall vascular development. We hope that we have addressed all the reviewer's concerns.

As to the in vivo relevance of JBL in different morphogenetic context and other species, we are addressing these issues in the future, for example the role of JBL in flow-mediated cell shape changes or blood vessel regression. To investigate JBL in mouse angiogenesis seems at this point difficult because of the lack of suitable in vivo life-imaging technology. As mentioned in our results, JBL are best imaged in single cell contacts (i.e. in junctional rings) rather than in multi-cellular configurations. In the mouse retina, for example, most junctional immunostainings focus either on multi-cellular situations, which makes JBL localization difficult or are not done at sufficient resolution. A pleasant exception is the paper by Franco et al. (PLoS Biology, 2015, Figure 2), which shows single junctional rings during retinal vascular remodelling. The junctional thickenings observed here are entirely consistent with the ones we describe in the zebrafish DLAV.

Mino points:

Typo: Supple Figure 1C, the Y-axis legend says: ...inflated... should probably be inflated I assume?

Typo: 318 prominent stress fibers during in this process

Reviewers' comments:

Reviewer #1 (Remarks to the Author):

The authors have improved their manuscript. Yet, several points have not been thoroughly experimentally addressed. In particular:

- The answer to Q1.1. led to the removal of some interesting statements/conclusions.
- The answer to Q4.1. is incomplete since no quantification of the results is shown.
- The answer to Q4.2. is very confusing. While the authors mentioned that they have added data points in the wt conditions, the graph shows very little or no additional data points. Furthermore no statistical analysis sustains the conclusion of the authors on the difference between wt and homozygote mutant conditions.
- The answer to Q4.3. is also very confusing. I still don't understand why the authors cannot perform the experiments with UCHD-GFP upon inhibition of Rac to compare the rac and ve-Cad phenotypes on the same graph. This would be important.
- Q4.4. It is in my view an important point that has not been addressed to support the general relevance of the study.

Reviewer #3 (Remarks to the Author):

The authors have done a nice job revising their paper and responding to my comments and those of the other reviewers. I have no further questions or comments.

Response to reviewer comments

Reviewers' comments in plain font, author reply in green

We thank the reviewers for their efforts to review our manuscript. While Reviewer#3 is giving his go ahead, Reviewer#2 has maintained some concerns. Below, please find our point-by-point responses.

The answer to Q1.1. led to the removal of some interesting statements/conclusions.

The reviewer is correct. As pointed out in our previous rebuttal, we have tried to quantify the spatiotemporal differences between “host” and “donor” F-actin pools. However, this proved to be impossible with the available (even though “state-of-the-art”) technology. We have toned down our conclusions accordingly.

The answer to Q4.1. is incomplete since no quantification of the results is shown.

We have now performed quantification of the spatial distribution of VE-cad relative to another junctional protein (ZO-1) (Pearson correlation coefficients of colocalization analysis). This analysis shows no significant difference in the localization of both proteins ($p=0.7498$). The quantification has been included in **Figure S4E**.

The answer to Q4.2. is very confusing. While the authors mentioned that they have added data points in the wt conditions, the graph shows very little or no additional data points. Furthermore no statistical analysis sustains the conclusion of the authors on the difference between wt and homozygote mutant conditions.

The reviewer is correct. During the preparation of the figures for the revision, we have accidentally used the old panel instead of the new dataset. In the new dataset there are significantly more measurements (old $n=11$, new $n=43$) and the p-value between wild-type and *ve-cad*^{ubs25} is highly significant ($p<0.001$). In the corrected version of the manuscript this is Figure 7c.

The answer to Q4.3. is also very confusing. I still don't understand why the authors cannot perform the experiments with UCHD-GFP upon inhibition of Rac to compare the rac and ve-Cad phenotypes on the same graph. This would be important.

We are not convinced that these additional experiments are justified. For one: these experiments were not originally requested by reviewer#1, who had stated ...

“ Q4.3: The authors need to show and to compare the JBL duration in VE-Cad mutant and upon Rac inhibition in the same graph.”

By now requesting these additional experiments, reviewer #1 is raising the bar late in the submission process, causing – in our opinion - an unnecessary delay in the publication of our work. One has to keep in mind that *in vivo* time-lapse imaging - at

the spatial and temporal resolution required for these experiments - is unique in its phenotypic depth and everything but trivial. Perhaps more importantly, even if we did perform this experiment, in our opinion it would add limited, if any, value to the study overall.

In Figure 2, we show a detailed analysis comparing the dynamics of JBLs based on both markers - EGFP-UCHD and VE-cad-Venus. Using either marker yielded almost identical results, with respect to JBL duration (Figure 2E) as well as orientation (Figure 2 F). This clearly indicated that both markers are labelling equivalent structures and are comparable for reading out JBL phenotypic changes.

Key panels from Figure 2 comparing duration (E) and orientation (F) of JBL during blood vessel anastomosis.

These observations led us to conclude that both markers can be used interchangeably, with respect to JBL dynamics. Differences in VE-cad and UCHD localization only became evident at higher spatial resolution (compare Figure 5B - 120 sec).

Key panels from Figure 5 showing differential distribution of VE-cadherin and F-actin. From left to right: merged channels//VE-cad-Venus//EGFP-UCHD//Schematic

However, these differences are very subtle and are not relevant, when considering general JBL dynamics at a very different scale (as we do in Figures 6 and 7 which the reviewer has queried). We did, however, notice that VE-cad-Venus is easier to use as it facilitates detection of endothelial cell junctions. While we prefer VE-cad-Venus as a superior marker over the EGFP-UCHD, it is crucial to note that it is not possible to use the VE-cad-Venus marker in the context of a VE-cad mutant embryo because this transgene completely rescues the VE-cad-Venus phenotype (as we have published previously in Lagendijk et al., 2017).

We have previously pointed this out - (as stated in our rebuttal: ... *“As a key issue, we have used different transgenic lines; although VE-cad-Venus transgene is superior to UCHD-*

GFP in unambiguously defining cell-cell junctions during anastomosis, it would have rescued any defects caused by ubs25 mutation.”)

In light of this and our extensive published data as well as data in the current study, it is reasonable to consider that VE-cad-Venus and EGFP-UCHD are equivalent markers for JBL dynamics. Indeed, in his original comment reviewer#1 seemed primarily concerned that we hadn't assembled our data within one graph (see above). As we pointed out in our rebuttal, a direct comparison between these data may mis-represent the specific experimental design that we used and give the reader the wrong impression. Our conclusions are completely supported by these two different experiments that utilize different but equivalent markers, however putting these two distinct experiments on one graph – we feel – would not be accurate.

Our work is primarily about the discovery of a very specific cellular activity, i.e. JBL. Using cutting edge *in vivo* time-lapse imaging we describe JBL dynamics during blood vessel morphogenesis. We show that relevant proteins VE-cad, ZO-1 and F-actin are dynamically and differently localized, leading to our model of JBL function. In addition, we show that F-actin dynamics (Rac1 inhibition) and VE-cadherin (ubs25 mutant) are functionally required for JBL dynamics. We feel that our new model of JBL function represents a considerable conceptual advance in the field. However, pin-pointing the exact regulation of JBL oscillations will require considerable future work and differentiating between the precise role Rac1 activity at the level of F-actin/VE-cadherin interaction will require future study.

Q4.4. It is in my view an important point that has not been addressed to support the general relevance of the study.

As we have outlined in our previous response it is not possible for us to directly score occurrence vs. non-occurrence of JBL *in vivo*. This is due to the experimental set-up, which requires us to perform life-imaging in small subregions of cell junctions - thus preventing a global view. However, we have put some thought into this question and have compared additional aspects of JBL dynamics in wild-types and *ve-cad* mutants. In addition to JBL duration (corrected Figure 7c), we have also quantified the lag phase between the formation of two JBL. This analysis shows that the intervals between JBL is prolonged in *ve-cad* mutants ($p < 0.001$), which effectively translates into the formation of fewer JBL over time in mutants. These findings are presented in **Figure S4F**.